# MALI: A MEMORY EFFICIENT AND REVERSE ACCURATE INTEGRATOR FOR NEURAL ODES

**Juntang Zhuang; Nicha C. Dvornek; Sekhar Tatikonda; James S. Duncan**
{j.zhuang; nicha.dvornek; sekhar.tatikonda; james.duncan} @yale.edu
Yale University, New Haven, CT, USA

## ABSTRACT

Neural ordinary differential equations (Neural ODEs) are a new family of deep-learning models with continuous depth. However, the numerical estimation of the gradient in the continuous case is not well solved: existing implementations of the adjoint method suffer from inaccuracy in reverse-time trajectory, while the naive method and the adaptive checkpoint adjoint method (ACA) have a memory cost that grows with integration time. In this project, based on the asynchronous leapfrog (ALF) solver, we propose the Memory-efficient ALF Integrator (MALI), which has a constant memory cost *w.r.t* number of solver steps in integration similar to the adjoint method, and guarantees accuracy in reverse-time trajectory (hence accuracy in gradient estimation). We validate MALI in various tasks: on image recognition tasks, to our knowledge, MALI is the first to enable feasible training of a Neural ODE on ImageNet and outperform a well-tuned ResNet, while existing methods fail due to either heavy memory burden or inaccuracy; for time series modeling, MALI significantly outperforms the adjoint method; and for continuous generative models, MALI achieves new state-of-the-art performance.We provide a pypi package: `https://jzkay12.github.io/TorchDiffEqPack`

## 1 INTRODUCTION

Recent research builds the connection between continuous models and neural networks. The theory of dynamical systems has been applied to analyze the properties of neural networks or guide the design of networks (Weinan, 2017; Ruthotto & Haber, 2019; Lu et al., 2018). In these works, a residual block (He et al., 2016) is typically viewed as a one-step Euler discretization of an ODE; instead of directly analyzing the discretized neural network, it might be easier to analyze the ODE.

Another direction is the neural ordinary differential equation (Neural ODE) (Chen et al., 2018), which takes a continuous depth instead of discretized depth. The dynamics of a Neural ODE is typically approximated by numerical integration with adaptive ODE solvers. Neural ODEs have been applied in irregularly sampled time-series (Rubanova et al., 2019), free-form continuous generative models (Grathwohl et al., 2018; Finlay et al., 2020), mean-field games (Ruthotto et al., 2020), stochastic differential equations (Li et al., 2020) and physically informed modeling (Sanchez-Gonzalez et al., 2019; Zhong et al., 2019).

Though the Neural ODE has been widely applied in practice, how to train it is not extensively studied. The *naive method* directly backpropagates through an ODE solver, but tracking a continuous trajectory requires a huge memory. Chen et al. (2018) proposed to use the *adjoint method* to determine the gradient in continuous cases, which achieves constant memory cost *w.r.t* integration time; however, as pointed out by Zhuang et al. (2020), the adjoint method suffers from numerical errors due to the inaccuracy in reverse-time trajectory. Zhuang et al. (2020) proposed the *adaptive checkpoint adjoint (ACA)* method to achieve accuracy in gradient estimation at a much smaller memory cost compared to the naive method, yet the memory consumption of ACA still grows linearly with integration time. Due to the non-constant memory cost, neither ACA nor naive method are suitable for large scale datasets (e.g. ImageNet) or high-dimensional Neural ODEs (e.g. FFJORD (Grathwohl et al., 2018)).

In this project, we propose the Memory-efficient Asynchronous Leapfrog Integrator (MALI) to achieve advantages of both the adjoint method and ACA: constant memory cost *w.r.t* integration time and accuracy in reverse-time trajectory. MALI is based on the asynchronous leapfrog (ALF) integrator (Mutze, 2013). With the ALF integrator, each numerical step forward in time is reversible. Therefore, with MALI, we delete the trajectory and only keep the end-time states, hence achieve constant memory cost *w.r.t* integration time; using the reversibility, we can accurately reconstruct the trajectory from the end-time value, hence achieve accuracy in gradient. Our contributions are:

1. We propose a new method (MALI) to solve Neural ODEs, which achieves constant memory cost *w.r.t* number of solver steps in integration and accuracy in gradient estimation. We provide theoretical analysis.

2. We validate our method with extensive experiments: (a) for image classification tasks, MALI enables a Neural ODE to achieve better accuracy than a well-tuned ResNet with the same number of parameters; to our knowledge, MALI is the first method to enable training of Neural ODEs on a large-scale dataset such as ImageNet, while existing methods fail due to either heavy memory burden or inaccuracy. (b) In time-series modeling, MALI achieves comparable or better results than other methods. (c) For generative modeling, a FFJORD model trained with MALI achieves new state-of-the-art results on MNIST and Cifar10.

## 2 PRELIMINARIES

### 2.1 NUMERICAL INTEGRATION METHODS

An ordinary differential equation (ODE) typically takes the form

$$\frac{\mathrm{d}z(t)}{\mathrm{d}t} = f_\theta(t, z(t)) \quad s.t. \quad z(t_0) = x, \ t \in [t_0, T], \quad Loss = L(z(T), y) \tag{1}$$

where $z(t)$ is the hidden state evolving with time, $T$ is the end time, $t_0$ is the start time (typically 0), $x$ is the initial state. The derivative of $z(t)$ w.r.t $t$ is defined by a function $f$, and $f$ is defined as a sequence of layers parameterized by $\theta$. The loss function is $L(z(T), y)$, where $y$ is the target variable. Eq. 1 is called the initial value problem (IVP) because only $z(t_0)$ is specified.

**Notations** We summarize the notations following Zhuang et al. (2020).

- $z_i(t_i)/\bar{z}(\tau_i)$: hidden state in forward/reverse time trajectory at time $t_i/\tau_i$.
- $\psi_h(t_i, z_i)$: the *numerical* solution at time $t_i + h$, starting from $(t_i, z_i)$ with a stepsize $h$.
- $N_f, N_z$: $N_f$ is the number of layers in $f$ in Eq. 1, $N_z$ is the dimension of $z$.
- $N_t/N_r$: number of discretized points (outer iterations in Algo. 1) in forward / reverse integration.
- $m$: average number of inner iterations in Algo. 1 to find an acceptable stepsize.

**Algorithm 1:** Numerical Integration

**Input** initial state $x$, start time $t_0$, end time $T$, error tolerance $etol$, initial stepsize $h$.
**Initialize** $z(0) = x, t = t_0$
**While** $t < T$
    $error\_est = \infty$
    **While** $error\_est > etol$
        $h \leftarrow h \times DecayFactor$
        $\hat{z}, error\_est = \psi_h(t, z)$
    **If** $error\_est < etol$
        $h \leftarrow h \times IncreaseFactor$
    $t \leftarrow t + h, \ z \leftarrow \hat{z}$

**Numerical Integration** The algorithm for general adaptive-stepsize numerical ODE solvers is summarized in Algo. 1 (Wanner & Hairer, 1996). The solver repeatedly advances in time by a step, which is the outer loop in Algo. 1 (blue curve in Fig. 1). For each step, the solver decreases the stepsize until the estimate of error is lower than the tolerance, which is the inner loop in Algo. 1 (green curve in Fig. 1). For fixed-stepsize solvers, the inner loop is replaced with a single evaluation of $\psi_h(t, z)$ using predefined stepsize $h$. Different methods typically use different $\psi$, for example different orders of the Runge-Kutta method (Runge, 1895).

### 2.2 ANALYTICAL FORM OF GRADIENT IN CONTINUOUS CASE

We first briefly introduce the analytical form of the gradient in the continuous case, then we compare different numerical implementations in the literature to estimate the gradient. The analytical form

Table 1: Comparison between different methods for gradient estimation in continuous case. MALI achieves reverse accuracy, constant memory *w.r.t* number of solver steps in integration, shallow computation graph and low computation cost.

| | Naive | Adjoint | ACA | MALI |
|---|---|---|---|---|
| Computation | $NzN_f \times N_t \times m \times 2$ | $NzN_f \times (N_t + N_r) \times m$ | $NzN_f \times N_t \times (m+1)$ | $NzN_f \times N_t \times (m+2)$ |
| Memory | $NzN_f \times N_t \times m$ | $NzN_f$ | $Nz(N_f + N_t)$ | $Nz(N_f + 1)$ |
| Computation graph depth | $N_f \times N_t \times m$ | $N_f \times N_r$ | $N_f \times N_t$ | $N_f \times N_t$ |
| Reverse accuracy | ✓ | ✗ | ✓ | ✓ |

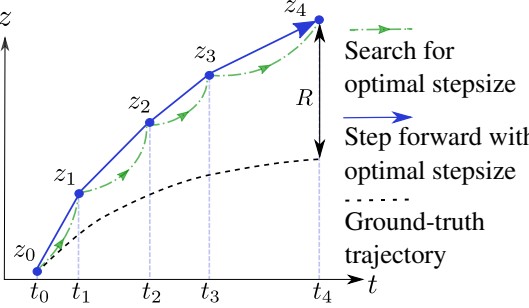

Figure 1: Illustration of numerical solver in forward-pass. For adaptive solvers, for each step forward-in-time, the stepsize is recursively adjusted until the estimated error is below predefined tolerance; the search process is represented by green curve, and the accepted step (ignore the search process) is represented by blue curve.

Figure 2: In backward-pass, the adjoint method reconstructs trajectory as a separate IVP. Naive, ACA and MALI track the forward-time trajectory, hence are accurate. ACA and MALI only backpropagate through the accepted step, while naive method backpropagates through the search process hence has deeper computation graphs.

of the gradient in the continuous case is

$$\frac{dL}{d\theta} = -\int_T^0 a(t)^\top \frac{\partial f(z(t), t, \theta)}{\partial \theta} dt \quad (2)$$

$$\frac{da(t)}{dt} + \Big(\frac{\partial f(z(t), t, \theta)}{\partial z(t)}\Big)^\top a(t) = 0 \ \ \forall t \in (0, T), \quad a(T) = \frac{\partial L}{\partial z(T)} \quad (3)$$

where $a(t)$ is the "adjoint state". Detailed proof is given in (Pontryagin, 1962). In the next section we compare different numerical implementations of this analytical form.

### 2.3 NUMERICAL IMPLEMENTATIONS IN THE LITERATURE FOR THE ANALYTICAL FORM

We compare different numerical implementations of the analytical form in this section. The forward-pass and backward-pass of different methods are demonstrated in Fig. 1 and Fig. 2 respectively. Forward-pass is similar for different methods. The comparison of backward-pass among different methods are summarized in Table. 1. We explain methods in the literature below.

**Naive method** The naive method saves all of the computation graph (including search for optimal stepsize, green curve in Fig. 2) in memory, and backpropagates through it. Hence the memory cost is $N_z N_f \times N_t \times m$ and depth of computation graph are $N_f \times N_t \times m$, and the computation is doubled considering both forward and backward passes. Besides the large memory and computation, the deep computation graph might cause vanishing or exploding gradient (Pascanu et al., 2013).

**Adjoint method** Note that we use "*adjoint state equation*" to refer to the *analytical form* in Eq. 2 and 3, while we use "*adjoint method*" to refer to the *numerical implementation* by Chen et al. (2018). As in Fig. 1 and 2, the adjoint method forgets forward-time trajectory (blue curve) to achieve memory cost $N_z N_f$ which is constant to integration time; it takes the end-time state (derived from forward-time integration) as the initial state, and solves a separate IVP (red curve) in reverse-time.

**Theorem 2.1.** *(Zhuang et al., 2020) For an ODE solver of order $p$, the error of the reconstructed initial value by the adjoint method is $\sum_{k=0}^{N-1} \big[ h_k^{p+1} D\Phi_{t_k}^T(z_k) l(t_k, z_k) + (-h_k)^{p+1} D\Phi_T^{t_k}(\overline{z_k}) \overline{l(t_k, \overline{z_k})} \big] + O(h^{p+1})$, where $\Phi$ is the ideal solution, $D\Phi$ is the Jacobian of $\Phi$, $l(t, z)$ and $\overline{l(t, z)}$ are the local error in forward-time and reverse-time integration respectively.*

Theorem 2.1 is stated as Theorem 3.2 in Zhuang et al. (2020); please see reference paper for detailed proof. To summarize, due to inevitable errors with numerical ODE solvers, the reverse-time trajectory (red curve, $\overline{z}(\tau)$) cannot match the forward-time trajectory (blue curve, $z(t)$) accurately. The error in $\overline{z}$ propagates to $\frac{\mathrm{d}L}{\mathrm{d}\theta}$ by Eq. 2, hence affects the accuracy in gradient estimation.

**Adaptive checkpoint adjoint (ACA)** To solve the inaccuracy of adjoint method, Zhuang et al. (2020) proposed ACA: ACA stores forward-time trajectory in memory for backward-pass, hence guarantees accuracy; ACA deletes the search process (green curve in Fig. 2), and only back-propagates through the accepted step (blue curve in Fig. 2), hence has a shallower computation graph ($N_f \times N_t$ for ACA vs $N_f \times N_t \times m$ for naive method). ACA only stores $\{z(t_i)\}_{i=1}^{N_t}$, and deletes the computation graph for $\{f(z(t_i), t_i)\}_{i=1}^{N_t}$, hence the memory cost is $N_z(N_f + N_t)$. Though the memory cost is much smaller than the naive method, it grows linearly with $N_t$, and can not handle very high dimensional models. In the following sections, we propose a method to overcome all these disadvantages of existing methods.

# 3 METHODS

## 3.1 ASYNCHRONOUS LEAPFROG INTEGRATOR

In this section we give a brief introduction to the asynchronous leapfrog (ALF) method (Mutze, 2013), and we provide theoretical analysis which is missing in Mutze (2013). For general first-order ODEs in the form of Eq. 1, the tuple $(z, t)$ is sufficient for most ODE solvers to take a step numerically. For ALF, the required tuple is $(z, v, t)$, where $v$ is the "approximated derivative". Most numerical ODE solvers such as the Runge-Kutta method (Runge, 1895) track state $z$ evolving with time, while ALF tracks the "augmented state" $(z, v)$. We explain the details of ALF as below.

| **Algorithm 2:** Forward of $\psi$ in ALF |
|---|
| **Input** $(z_{in}, v_{in}, s_{in}, h)$ where $s_{in}$ is current time, $z_{in}$ and $v_{in}$ are correponding values at time $s_{in}$, $h$ is stepsize. |
| **Forward** $\quad s_1 = s_{in} + h/2$ 
 $\qquad\qquad k_1 = z_{in} + v_{in} \times h/2$ 
 $\qquad\qquad u_1 = f(k_1, s_1)$ 
 $\qquad\qquad v_{out} = v_{in} + 2(u_1 - v_{in})$ 
 $\qquad\qquad z_{out} = k_1 + v_{out} \times h/2$ 
 $\qquad\qquad s_{out} = s_1 + h/2$ |
| **Output** $\quad (z_{out}, v_{out}, s_{out}, h)$ |

| **Algorithm 3:** $\psi^{-1}$ (Inverse of $\psi$) in ALF |
|---|
| **Input** $(z_{out}, v_{out}, s_{out}, h)$ where $s_{out}$ is current time, $z_{out}$ and $v_{out}$ are corresponding values at $s_{out}$, $h$ is stepsize. |
| **Inverse** $\quad s_1 = s_{out} - h/2$ 
 $\qquad\qquad k_1 = z_{out} - v_{out} \times h/2$ 
 $\qquad\qquad u_1 = f(k_1, s_1)$ 
 $\qquad\qquad v_{in} = 2u_1 - v_{out}$ 
 $\qquad\qquad z_{in} = k_1 - v_{in} \times h/2$ 
 $\qquad\qquad s_{in} = s_1 - h/2$ |
| **Output** $\quad (z_{in}, v_{in}, s_{in}, h)$ |

**Procedure of ALF** Different ODE solvers have different $\psi$ in Algo. 1, hence we only summarize $\psi$ for ALF in Algo. 2. Note that for a complete algorithm of integration for ALF, we need to plug Algo. 2 into Algo. 1. The forward-pass is summarized in Algo. 2. Given stepsize $h$, with input $(z_{in}, v_{in}, s_{in})$, a single step of ALF outputs $(z_{out}, v_{out}, s_{out})$.

As in Fig. 3, given $(z_0, v_0, t_0)$, the numerical forward-time integration calls Algo. 2 iteratively:

Figure 3: With ALF method, given any tuple $(z_j, v_j, t_j)$ and discretized time points $\{t_i\}_{i=1}^{N_t}$, we can reconstruct the entire trajectory accurately due to the reversibility of ALF.

$$(z_i, v_i, t_i, h_i) = \psi(z_{i-1}, v_{i-1}, t_{i-1}, h_i)$$
$$s.t. \; h_i = t_i - t_{i-1}, \; i = 1, 2, ...N_t \quad (4)$$

**Invertibility of ALF** An interesting property of ALF is that $\psi$ defines a bijective mapping; therefore, we can reconstruct $(z_{in}, v_{in}, s_{in}, h)$ from $(z_{out}, v_{out}, s_{out}, h)$, as demonstrated in Algo. 7. As in Fig. 3, we can reconstruct the entire trajectory given the state $(z_j, v_j)$ at time $t_j$, and the discretized time points $\{t_0, ...t_{N_t}\}$. For example, given $(z_{N_t}, v_{N_t})$ and $\{t_i\}_{i=0}^{N_t}$, the trajectory for Eq. 4 is reconstructed:

$$(z_{i-1}, v_{i-1}, t_{i-1}, h_i) = \psi^{-1}(z_i, v_i, t_i, h_i) \; s.t. \; h_i = t_i - t_{i-1}, \; i = N_t, N_t - 1, ..., 1 \quad (5)$$

In the following sections, we will show the invertibility of ALF is the key to maintain accuracy at a constant memory cost to train Neural ODEs. Note that "inverse" refers to reconstructing the input from the output without computing the gradient, hence is different from "back-propagation".

**Initial value** For an initial value problem (IVP) such as Eq. 1, typically $z_0 = z(t_0)$ is given while $v_0$ is undetermined. We can construct $v_0 = f(z(t_0), t_0)$, so the initial augmented state is $(z_0, v_0)$.

**Difference from midpoint integrator** The midpoint integrator (Süli & Mayers, 2003) is similar to Algo. 2, except that it recomputes $v_{in} = f(z_{in}, s_{in})$ for every step, while ALF directly uses the input $v_{in}$. Therefore, the midpoint method does not have an explicit form of inverse.

**Local truncation error** Theorem 3.1 indicates that the local truncation error of ALF is of order $O(h^3)$; this implies the global error is $O(h^2)$. Detailed proof is in Appendix A.3.

**Theorem 3.1.** *For a single step in ALF with stepsize $h$, the local truncation error of $z$ is $O(h^3)$, and the local truncation error of $v$ is $O(h^2)$.*

**A-Stability** The ALF solver has a limited stability region, but this can be solved with damping. The damped ALF replaces the update of $v_{out}$ in Algo. 2 with $v_{out} = v_{in} + 2\eta(u_1 - v_{in})$, where $\eta$ is the "damping coefficient" between 0 and 1. We have the following theorem on its numerical stability.

**Theorem 3.2.** *For the damped ALF integrator with stepsize $h$, where $\sigma_i$ is the $i$-th eigenvalue of the Jacobian $\frac{\partial f}{\partial z}$, then the solver is A-stable if $\left| 1 + \eta(h\sigma_i - 1) \pm \sqrt{\eta\left[2h\sigma_i + \eta(h\sigma_i - 1)^2\right]} \right| < 1, \forall i$*

Proof is in Appendix A.4 and A.5. Theorem 3.2 implies the following: when $\eta = 1$, the damped ALF reduces to ALF, and the stability region is empty; when $0 < \eta < 1$, the stability region is non-empty. However, stability describes the behaviour when $T$ goes to infinity; in practice we always use a bounded $T$ and ALF performs well. Inverse of damped ALF is in Appendix A.5.

### 3.2 Memory-efficient ALF Integrator (MALI) for gradient estimation

An ideal solver for Neural ODEs should achieve two goals: accuracy in gradient estimation and constant memory cost *w.r.t* integration time. Yet none of the existing methods can achieve both goals. We propose a method based on the ALF solver, which to our knowledge is the first method to achieve the two goals simultaneously.

---

**Algorithm 4:** MALI to acheive accuracy at a constant memory cost *w.r.t* integration time

---

**Input** Initial state $z_0$, start time $t_0$, end time $T$
**Forward**
    Apply the numerical integration in Algo. 1, with the $\psi$ function defined by Algo. 2.
    Delete computation graph on the fly, only keep end-time state $(z_{N_t}, v_{N_t})$
    Keep *accepted* discretized time points $\{t_i\}_{i=0}^{N_t}$ (ignore process to search for optimal stepsize)
**Backward**
    Initialize $a(T) = \frac{\partial L}{\partial z(T)}$ by Eq. 3, initialize $\frac{dL}{d\theta} = 0$
    **For** $i$ in $\{N_t, N_t - 1, ..., 2, 1\}$:
        Reconstruct $(z_{i-1}, v_{i-1})$ from $(z_i, v_i)$ by Algo. 7
        Local forward $(z_i, v_i, t_i, h_i) = \psi(z_{i-1}, v_{i-1}, t_{i-1}, h_i)$
        Local backward, get $\frac{\partial f(z_{i-1}, t_{i-1}, \theta)}{\partial z_{i-1}}$ and $\frac{\partial f(z_{i-1}, t_{i-1}, \theta)}{\partial \theta}$
        Update $a(t)$ and $\frac{dL}{d\theta}$ by Eq. 2 and Eq. 3 discretized at time points $t_{i-1}$ and $t_i$
        Delete local computation graph
    **Output** the adjoint state $a(t_0)$ (gradient *w.r.t* input $z_0$) and parameter gradient $\frac{dL}{d\theta}$

---

**Procedure of MALI** Details of MALI are summarized in Algo. 4. For the forward-pass, we only keep the end-time state $(z_{N_t}, v_{N_t})$ and the *accepted* discretized time points (blue curves in Fig. 1 and 2). We ignore the search process for optimal stepsize (green curve in Fig. 1 and 2), and delete other variables to save memory. During the backward pass, we can reconstruct the forward-time trajectory as in Eq. 5, then calculate the gradient by numerical discretization of Eq. 2 and Eq. 3.

**Constant memory cost *w.r.t* number of solver steps in integration** We delete the computation graph and only keep the end-time state to save memory. The memory cost is $N_z(N_f + 1)$, where

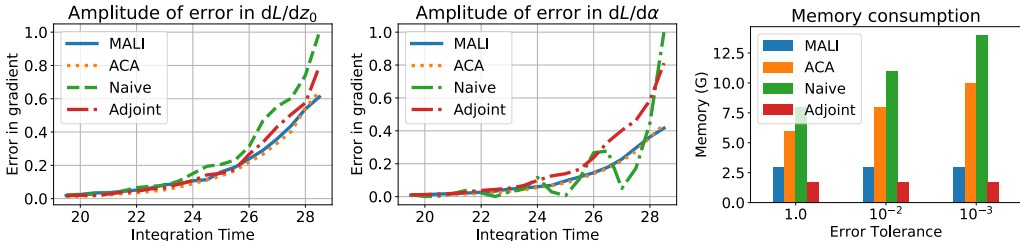

Figure 4: Comparison of error in gradient in Eq. 6. (a) error in $\frac{\mathrm{d}L}{\mathrm{d}z_0}$. (b) error in $\frac{\mathrm{d}L}{\mathrm{d}\alpha}$. (c) memory cost.

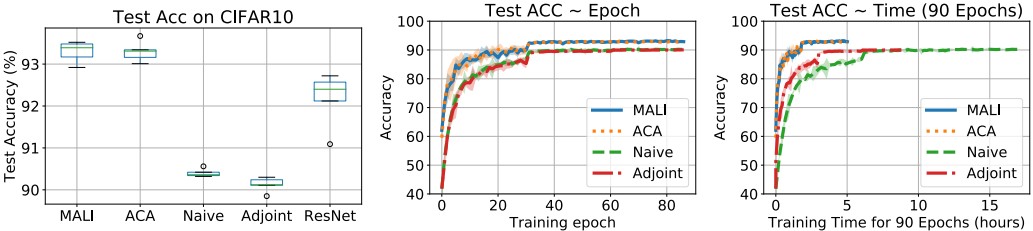

Figure 5: Results on Cifar10. From left to right: (1) box plot of test accuracy (first 4 columns are Neural ODEs, last is ResNet); (2) test accuracy $\pm std$ v.s. training epoch for Neural ODE; (3) test accuracy $\pm std$ v.s. training time of 90 epochs for Neural ODE.

$N_z N_f$ is due to evaluating $f(z,t)$ and is irreducible for all methods. Compared with the adjoint method, MALI only requires extra $N_z$ memory to record $v_{N_t}$, and also has a constant memory cost *w.r.t* time step $N_t$. The memory cost is $N_z(N_f + 1)$.

**Accuracy** Our method guarantees the accuracy of reverse-time trajectory (e.g. blue curve in Fig. 2 matches the blue curve in Fig. 1), because ALF is explicitly invertible for free-form $f$ (see Algo. 7). Therefore, the gradient estimation in MALI is more accurate compared to the adjoint method.

**Computation cost** Recall that on average it takes $m$ steps to find an acceptable stepsize, whose error estimate is below tolerance. Therefore, the forward-pass with search process has computation burden $N_z \times N_f \times N_t \times m$. Note that we only reconstruct and backprop through the *accepted* step and ignore the search process, hence it takes another $N_z \times N_f \times N_t \times 2$ computation. The overall computation burden is $N_z N_f \times N_t \times (m + 2)$ as in Table 1.

**Shallow computation graph** Similar to ACA, MALI only backpropagates through the accepted step (blue curve in Fig. 2) and ignores the search process (green curve in Fig. 2), hence the depth of computation graph is $N_f \times N_t$. The computation graph of MALI is much shallower than the naive method, hence is more robust to vanishing and exploding gradients (Pascanu et al., 2013).

**Summary** The adjoint method suffers from inaccuracy in reverse-time trajectory, the naive method suffers from exploding or vanishing gradient caused by deep computation graph, and ACA finds a balance but the memory grows linearly with integration time. MALI achieves accuracy in reverse-time trajectory, constant memory *w.r.t* integration time, and a shallow computation graph.

## 4 EXPERIMENTS

### 4.1 VALIDATION ON A TOY EXAMPLE

We compare the performance of different methods on a toy example, defined as

$$L(z(T)) = z(T)^2 \ \ s.t. \ \ z(0) = z_0, \ \ \mathrm{d}z(t)/\mathrm{d}t = \alpha z(t) \tag{6}$$

The analytical solution is

$$z(t) = z_0 e^{\alpha t}, \ \ L = z_0^2 e^{2\alpha T}, \ \ \mathrm{d}L/\mathrm{d}z_0 = 2z_0 e^{2\alpha T}, \ \ \mathrm{d}L/d\alpha = 2T z_0^2 e^{2\alpha T} \tag{7}$$

We plot the amplitude of error between numerical solution and analytical solution varying with $T$ (integrated under the same error tolerance, rtol $= 10^{-5}$, atol $= 10^{-6}$) in Fig 4. ACA and MALI have similar errors, both outperforming other methods. We also plot the memory consumption for

Table 2: Top-1 test accuracy of Neural ODE and ResNet on ImageNet. Neural ODE is trained with MALI, and ResNet is trained as the original model; Neural ODE is tested using different solvers *without* retraining.

| | | Fixed-stepsize solvers of various stepsizes | | | | | Adaptive-stepsize solver of various tolerances | | | |
|---|---|---|---|---|---|---|---|---|---|---|
| | Stepsize | 1 | 0.5 | 0.25 | 0.15 | 0.1 | Tolerance | 1.00E+00 | 1.00E-01 | 1.00E-02 |
| Neural ODE | MALI | 42.33 | 66.4 | 69.59 | 70.17 | 69.94 | MALI | 62.56 | 69.89 | 69.87 |
| | Euler | 21.94 | 61.25 | 67.38 | 68.69 | 70.02 | Heun-Euler | 68.48 | 69.87 | 69.88 |
| | RK2 | 42.33 | 69 | 69.72 | 70.14 | 69.92 | RK23 | 50.77 | 69.89 | 69.93 |
| | RK4 | 12.6 | 69.99 | 69.91 | **70.21** | 69.96 | Dopri5 | 52.3 | 68.58 | 69.71 |
| ResNet | | 70.09 | | | | | | | | |

Table 3: Top-1 accuracy under FGSM attack. $\epsilon$ is the perturbation amplitude. For Neural ODE models, row names represent the solvers to derive the gradient for attack, and column names represent solvers for inference on the perturbed image.

| | | $\epsilon = 1/255$ | | | | $\epsilon = 2/255$ | | | |
|---|---|---|---|---|---|---|---|---|---|
| | | MALI | Heun-Euler | RK23 | Dopri5 | MALI | Heun-Euler | RK23 | Dopri5 |
| Neural ODE | MALI | 14.69 | 14.72 | 14.77 | 15.71 | 10.38 | 10.46 | 10.62 | 10.62 |
| | Heun-Euler | 14.77 | 14.75 | 14.80 | **15.74** | 10.63 | 10.47 | 10.44 | 10.49 |
| | RK23 | 14.82 | 14.77 | 14.79 | 15.69 | **10.78** | 10.53 | 10.48 | 10.56 |
| | Dopri5 | 14.82 | 14.78 | 14.79 | 15.15 | 10.76 | 10.49 | 10.48 | 10.51 |
| ResNet | | 13.02 | | | | 9.57 | | | |

different methods on a Neural ODE with the same input in Fig. 4. As the error tolerance decreases, the solver evaluates more steps, hence the naive method and ACA increase memory consumption, while MALI and the adjoint method have a constant memory cost. These results validate our analysis in Sec. 3.2 and Table 1, and shows MALI achieves accuracy at a constant memory cost.

## 4.2 IMAGE RECOGNITION WITH NEURAL ODE

We validate MALI on image recognition tasks using Cifar10 and ImageNet datasets. Similar to Zhuang et al. (2020), we modify a ResNet18 into its corresponding Neural ODE: the forward function is $y = x + f_\theta(x)$ and $y = x + \int_0^T f_\theta(z)\mathrm{d}t$ for the residual block and Neural ODE respectively, where the same $f_\theta$ is shared. We compare MALI with the naive method, adjoint method and ACA.

**Results on Cifar10** Results of 5 independent runs on Cifar10 are summarized in Fig. 5. MALI achieves comparable accuracy to ACA, and both significantly outperform the naive and the adjoint method. Furthermore, the training speed of MALI is similar to ACA, and both are almost two times faster than the adjoint memthod, and three times faster than the naive method. This validates our analysis on accuracy and computation burden in Table 1.

**Accuracy on ImageNet** Due to the heavy memory burden caused by large images, the naive method and ACA are unable to train a Neural ODE on ImageNet with 4 GPUs; only MALI and the adjoint method are feasible due to the constant memory. We also compare the Neural ODE to a standard ResNet. As shown in Fig. 6, the accuracy of the Neural ODE trained with MALI closely follows ResNet, and significantly outperforms the adjoint method (top-1 validation: 70% v.s. 63%).

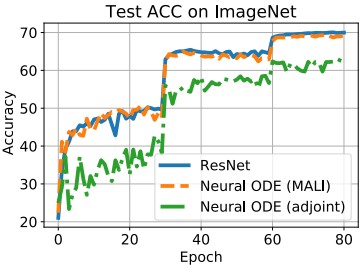

Figure 6: Top-1 accuracy on ImageNet validation dataset.

**Invariance to discretization scheme** A continuous model should be invariant to discretization schemes (e.g. different types of ODE solvers) as long as the discretization is sufficiently accurate. We test the Neural ODE using different solvers *without* re-training; since ResNet is often viewed as a one-step Euler discretization of an ODE (Haber & Ruthotto, 2017), we perform similar experiments. As shown in Table 2, Neural ODE consistently achieves high accuracy ($\sim$70%), while ResNet drops to random guessing ($\sim$0.1%) because ResNet as a one-step Euler discretization fails to be a meaningful dynamical system (Queiruga et al., 2020).

**Robustness to adversarial attack** Hanshu et al. (2019) demonstrated that Neural ODE is more robust to adversarial attack than ResNet on small-scale datasets such as Cifar10. We validate this result on the large-scale ImageNet dataset. The top-1 accuracy of Neural ODE and ResNet under FGSM attack (Goodfellow et al., 2014) are summarized in Table 3. For Neural ODE, due to its invariance to discretization scheme, we derive the gradient for attack using a certain solver (row in

Table 4: Test MSE ($\times 0.01$) on Mujoco dataset (lower is better). Results marked with superscript numbers correspond to literature in the footnote.

| Percentage of training data | RNN[1] | RNN-GRU[1] | Latent-ODE | | | |
|---|---|---|---|---|---|---|
| | | | Adjoint[1] | Naive[2] | ACA[2] | MALI |
| 10% | 2.45[1] | 1.97[2] | 0.47[1] | 0.36[2] | **0.31**[2] | 0.35 |
| 20% | 1.71[1] | 1.42[1] | 0.44[1] | 0.30[2] | **0.27**[2] | **0.27** |
| 50% | 0.79[1] | 0.75[1] | 0.40[1] | 0.29[2] | **0.26**[2] | **0.26** |

Table 5: Test ACC on Speech Command Dataset

| Method | Accuracy (%) |
|---|---|
| Adjoint[3] | $92.8 \pm 0.4$ |
| SemiNorm[3] | $92.9 \pm 0.4$ |
| Naive | $93.2 \pm 0.2$ |
| ACA | $93.2 \pm 0.2$ |
| MALI | $\mathbf{93.7 \pm 0.3}$ |

Table 3), and inference on the perturbed images using various solvers. For different combinations of solvers and perturbation amplitudes, Neural ODE consistently outperforms ResNet.

**Summary** In image recognition tasks, we demonstrate Neural ODE is accurate, invariant to discretization scheme, and more robust to adversarial attack than ResNet. Note that detailed explanation on the robustness of Neural ODE is out of the scope for this paper, but to our knowledge, MALI is the *first* method to enable training of Neural ODE on large datasets due to constant memory cost.

### 4.3 TIME-SERIES MODELING

We apply MALI to latent-ODE (Rubanova et al., 2019) and Neural Controlled Differential Equation (Neural CDE) (Kidger et al., 2020a;b). Our experiment is based on the official implementation from the literature. We report the mean squared error (MSE) on the *Mujoco* test set in Table 4, which is generated from the "Hopper" model using DeepMind control suite (Tassa et al., 2018); for all experiments with different ratios of training data, MALI achieves similar MSE to ACA, and both outperform the adjoint and naive method. We report the test accuracy on the *Speech Command* dataset for Neural CDE in Table 5; MALI achieves a higher accuracy than competing methods.

### 4.4 CONTINUOUS GENERATIVE MODELS

We apply MALI on FFJORD (Grathwohl et al., 2018), a free-from continuous generative model, and compare with several variants in the literature (Finlay et al., 2020; Kidger et al., 2020a). Our experiment is based on the official implementaion of Finlay et al. (2020); for a fair comparison, we train with MALI, and test with the same solver as in the literature (Grathwohl et al., 2018; Finlay et al., 2020), the *Dopri5* solver with rtol = atol = $10^{-5}$ from the *torchdiffeq* package (Chen et al., 2018). Bits per dim (BPD, lower is better) on validation set for various datasets are reported in Table 6. For continuous models, MALI consistently generates the lowest BPD, and outperforms the Vanilla FFJORD (trained with adjoint), RNODE (regularized FFJORD) and the SemiNorm Adjoint (Kidger et al., 2020a). Furthermore, FFJORD trained with MALI achieves comparable BPD to state-of-the-art discrete-layer flow models in the literature. Please see Sec. B.3 for generated samples.

## 5 RELATED WORKS

Besides ALF, the symplectic integrator (Verlet, 1967; Yoshida, 1990) is also able to reconstruct trajectory accurately, yet it's typically restricted to second order Hamiltonian systems (De Almeida, 1990), and are unsuitable for general ODEs. Besides aforementioned methods, there are other methods for gradient estimation such as interpolated adjoint (Daulbaev et al., 2020) and spectral method (Quaglino et al., 2019), yet the implementations are involved and not publicly available. Other works focus on the theoretical properties of Neural ODEs (Dupont et al., 2019; Tabuada & Gharesifard, 2020; Massaroli et al., 2020). Neural ODE is recently applied to stochastic differential equation (Li et al., 2020), jump differential equation (Jia & Benson, 2019) and auto-regressive models (Wehenkel & Louppe, 2019).

## 6 CONCLUSION

Based on the asynchronous leapfrog integrator, we propose MALI to estimate the gradient for Neural ODEs. To our knowledge, our method is the first to achieve accuracy, fast speed and a constant memory cost. We provide comprehensive theoretical analysis on its properties. We validate MALI

---

[0]1. Rubanova et al. (2019); 2. Zhuang et al. (2020); 3. Kidger et al. (2020a); 4. Chen et al. (2018); 5. Finlay et al. (2020); 6. Dinh et al. (2016); 7. Behrmann et al. (2019); 8. Kingma & Dhariwal (2018); 9. Ho et al. (2019); 10. Chen et al. (2019)

Table 6: Bits per dim (BPD) of generative models, *lower* is better. Results marked with superscript numbers correspond to literature in the footnote.

| Dataset | Continuous Flow (FFJORD) | | | | Discrete Flow | | | | |
|---|---|---|---|---|---|---|---|---|---|
| | Vanilla[4] | RNODE[5] | SemiNorm[3] | MALI | RealNVP[6] | i-ResNet[7] | Glow[8] | Flow++[9] | Residual Flow[10] |
| MNIST | 0.99[4] | 0.97[5] | 0.96[3] | **0.87** | 1.06[6] | 1.05[7] | 1.05[8] | - | 0.97[10] |
| CIFAR10 | 3.40[4] | 3.38[5] | 3.35[3] | **3.27** | 3.49[6] | 3.45[7] | 3.35[8] | 3.28[9] | 3.28[10] |
| ImageNet64 | - | 3.83[5] | - | **3.71** | 3.98[6] | - | 3.81[8] | - | 3.76[10] |

with extensive experiments, and achieved new state-of-the-art results in various tasks, including image recognition, continuous generative modeling, and time-series modeling.

## 7 ACKNOWLEDGEMENT

This research was funded by the National Institutes of Health (NINDS-R01NS035193)

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

CONTENTS (APPENDIX)

# A  THEORETICAL PROPERTIES OF ALF INTEGRATOR

## A.1  ALGORITHM OF ALF

For the ease of reading, we write the algorithm for $\psi$ in ALF below, which is the same as Algo. 2 in the main paper, but uses slightly different notations for the ease of analysis.

---

**Algorithm 1:** Forward of $\psi$ in ALF

---

**Input** $(\widehat{z_{in}}, \widehat{v_{in}}, s_{in}, h) = (\widehat{z}_0, \widehat{v}_0, s_0, h)$ where $s_0$ is current time, $\widehat{z}_0$ and $\widehat{v}_0$ are corresponding values at time $s_0$; stepsize $h$.

**Forward**

$$s_1 = s_0 + h/2 \tag{1}$$
$$\widehat{z}_1 = \widehat{z}_0 + \widehat{v}_0 \times h/2 \tag{2}$$
$$\widehat{v}_1 = f(\widehat{z}_1, s_1) \tag{3}$$
$$\widehat{v}_2 = \widehat{v}_1 + (\widehat{v}_1 - \widehat{v}_0) \tag{4}$$
$$\widehat{z}_2 = \widehat{z}_1 + \widehat{v}_2 \times h/2 \tag{5}$$
$$s_2 = s_1 + h/2 \tag{6}$$

**Output** $\qquad\qquad (\widehat{z_{out}}, \widehat{v_{out}}, s_{out}, h) = (\widehat{z}_2, \widehat{v}_2, s_2, h)$

---

For simplicity, we can re-write the forward of ALF as

$$\begin{bmatrix} \widehat{z}_2 \\ \widehat{v}_2 \end{bmatrix} = \begin{bmatrix} \widehat{z}_0 + h f(\widehat{z}_0 + \frac{h}{2}\widehat{v}_0, s_0 + \frac{h}{2}) \\ 2f(\widehat{z}_0 + \frac{h}{2}\widehat{v}_0, s_0 + \frac{h}{2}) - \widehat{v}_0 \end{bmatrix} \tag{7}$$

Similarly, the inverse of ALF can be written as

$$\begin{bmatrix} \widehat{z}_0 \\ \widehat{v}_0 \end{bmatrix} = \begin{bmatrix} \widehat{z}_2 - h f(\widehat{z}_2 - \frac{h}{2}\widehat{v}_2, s_2 - \frac{h}{2}) \\ 2f(\widehat{z}_2 - \frac{h}{2}\widehat{v}_2, s_2 - \frac{h}{2}) - \widehat{v}_2 \end{bmatrix} \tag{8}$$

## A.2  PRELIMINARIES

For an ODE of the form

$$\frac{\mathrm{d}z(t)}{\mathrm{d}t} = f(z(t), t) \tag{9}$$

We have:

$$\frac{\mathrm{d}^2 z(t)}{dt^2} = \frac{\mathrm{d}}{\mathrm{d}t} f(z(t), t) = \frac{\partial f(z(t), t)}{\partial t} + \frac{\partial f(z(t), t)}{\partial z} \frac{\mathrm{d}z(t)}{\mathrm{d}t} \tag{10}$$

For the ease of notation, we re-write Eq. 10 as

$$\frac{\mathrm{d}^2 z(t)}{dt^2} = f_t + f_z f \tag{11}$$

where $f_t$ and $f_z$ represents the partial derivative of $f$ w.r.t $t$ and $z$ respectively.

## A.3  LOCAL TRUNCATION ERROR OF ALF

**Theorem A.1** (Theorem 3.1 in the main paper). *For a single step in ALF with stepsize $h$, the local truncation error of $z$ is $O(h^3)$, and the local truncation errof of $v$ is $O(h^2)$.*

*Proof.* Under the same notation as Algo. 1, denote the ground-truth state of $z$ and $v$ starting from $(\widehat{z}_0, s_0)$ as $\widetilde{z}$ and $\widetilde{v}$ respectively. Then the local truncation error is

$$L_z = \widetilde{z}(s_0 + h) - \widehat{z}_2, \ \ L_v = \widetilde{v}(s_0 + h) - \widehat{v}_2 \tag{12}$$

We estimate $L_z$ and $L_v$ in terms of polynomial of $h$.

Under mild assumptions that $f$ is smooth up to 2nd order almost everywhere (this is typically satisfied with neural networks with bounded weights), hence Taylor expansion is meaningful for $f$. By Eq. 11, the Taylor expansion of $\widetilde{z}$ around point $(\widehat{z_0}, \widehat{v_0}, s_0)$ is

$$\widetilde{z}(s_0 + h) = \widehat{z_0} + h\frac{dz}{dt} + \frac{h^2}{2}\frac{d^2z}{dt^2} + O(h^3) \tag{13}$$

$$= \widehat{z_0} + hf(\widehat{z_0}, s_0) + \frac{h^2}{2}\Big(f_t(\widehat{z_0}, s_0) + f_z(\widehat{z_0}, s_0)f(\widehat{z_0}, s_0)\Big) + O(h^3) \tag{14}$$

Next, we analyze accuracy of the numerical approximation. For simplicity, we directly analyze Eq. 7 by performing Taylor Expansion on $f$.

$$f(\widehat{z_0} + \frac{h}{2}\widehat{v_0}, s_0 + \frac{h}{2}) = f(\widehat{z_0}, s_0) + \frac{h}{2}f_t(\widehat{z_0}, s_0) + \frac{h\widehat{v_0}}{2}f_z(\widehat{z_0}, s_0) + O(h^2) \tag{15}$$

$$\widehat{z_2} = \widehat{z_0} + hf(\widehat{z_0} + \frac{h}{2}\widehat{v_0}, s_0 + \frac{h}{2}) \tag{16}$$

Plug Eq. 14, Eq. 15 and E.q. 16 into the definition of $L_z$, we get

$$L_z = \widetilde{z}(s_0 + h) - \widehat{z_2} \tag{17}$$

$$= \Big[\widehat{z_0} + hf(\widehat{z_0}, s_0) + \frac{h^2}{2}\Big(f_t(\widehat{z_0}, s_0) + f_z(\widehat{z_0}, s_0)f(\widehat{z_0}, s_0)\Big)\Big]$$

$$- \Big[\widehat{z_0} + h\Big(f(\widehat{z_0}, s_0) + \frac{h}{2}f_t(\widehat{z_0}, s_0) + \frac{h\widehat{v_0}}{2}f_z(\widehat{z_0}, s_0)\Big)\Big] + O(h^3) \tag{18}$$

$$= \frac{h^2}{2}f_z(\widehat{z_0}, s_0)\Big(f(\widehat{z_0}, s_0) - \widehat{v_0}\Big) + O(h^3) \tag{19}$$

Therefore, if $\left|f(\widehat{z_0}, s_0) - \widehat{v_0}\right|$ is of order $O(1)$, $L_z$ is of order $O(h^2)$; if $\left|f(\widehat{z_0}, s_0) - \widehat{v_0}\right|$ is of order $O(h)$ or smaller, then $L_z$ is of order $O(h^3)$. Specifically, at the start time of integration, we have $\left|f(\widehat{z_0}, s_0) - \widehat{v_0} = 0\right|$, by induction, $L_z$ at end time is $O(h^3)$.

Next we analyze the local truncation error in $v$, denoted as $L_v$. Denote the ground truth as $\widetilde{v}(t_0 + h)$, we have

$$\widetilde{v}(s_0 + h) = f\big(\widetilde{z}(s_0 + h), s_0 + h\big) \tag{20}$$

$$= f(\widehat{z_0}, s_0) + hf_t(\widehat{z_0}, s_0) + \big(\widetilde{z}(s_0 + h) - \widehat{z_0}\big)f_z(\widehat{z_0}, s_0) + O(h^2) \tag{21}$$

Next we analyze the error in the numerical approximation. Plug Eq. 15 into Eq. 7,

$$\widehat{v_2} = 2f(\widehat{z_0} + \frac{h}{2}\widehat{v_0}, s_0 + \frac{h}{2}) - \widehat{v_0} \tag{22}$$

$$= f(\widehat{z_0}, s_0) + \big(f(\widehat{z_0}, s_0) - \widehat{v_0}\big) + hf_t(\widehat{z_0}, s_0) + h\widehat{v_0}f_z(\widehat{z_0}, s_0) + O(h^2) \tag{23}$$

From Eq. 14, Eq. 21 and Eq. 23, we have

$$L_v = \widetilde{v}(s_0 + h) - \widehat{v_2} \tag{24}$$

$$= \Big(f(\widehat{z_0}, s_0) - \widehat{v_0}\Big) + \Big(\widetilde{z}(s_0 + h) - \big(\widehat{z_0} + h\widehat{v_0}\big)\Big)f_z(\widehat{z_0}, s_0) + O(h^2) \tag{25}$$

$$= \Big(f(\widehat{z_0}, s_0) - \widehat{v_0}\Big) + h\Big(f(\widehat{z_0}, s_0) - \widehat{v_0}\Big)f_z(\widehat{z_0}, s_0) + O(h^2) \tag{26}$$

The last equation is derived by plugging in Eq. 14. Note that Eq. 26 holds for every single step forward in time, and at the start time of integration, we have $\left|f(\widehat{z_0}, s_0) - \widehat{v_0}\right| = 0$ due to our initialization as in Sec. 3.1 of the main paper. Therefore, by induction, $L_v$ is of order $O(h^2)$ for consecutive steps. □

## A.4 Stability Analysis

**Lemma A.1.1.** *For a matrix of the form* $\begin{bmatrix} A & B \\ C & D \end{bmatrix}$, *if* $A, B, C, D$ *are square matrices of the same shape, and* $CD = DC$, *then we have* $\det \begin{bmatrix} A & B \\ C & D \end{bmatrix} = \det(AD - BC)$

*Proof.* See (Silvester, 2000) for a detailed proof. □

**Theorem A.2.** *For ALF integrator with stepsize* $h$, *if* $h\sigma_i$ *is 0 or is imaginary with norm no larger than 1, where* $\sigma_i$ *is the* $i$-*th eigenvalue of the Jacobian* $\frac{\partial f}{\partial z}$, *then the solver is on the critical boundary of A-stability; otherwise, the solver is not A-stable.*

*Proof.* A solver is A-stable is equivalent to the eigenvalue of the numerical forward has a norm below 1. We calculate the eigenvalue of $\psi$ below.

For the function defined by Eq. 7, the Jacobian is

$$J = \begin{bmatrix} \frac{\partial \widehat{z_2}}{\partial z_0} & \frac{\partial \widehat{z_2}}{\partial \widehat{v_0}} \\ \frac{\partial \widehat{v_2}}{\partial z_0} & \frac{\partial \widehat{v_2}}{\partial \widehat{v_0}} \end{bmatrix} = \begin{bmatrix} I + h\frac{\partial f}{\partial z} & \frac{h^2}{2}\frac{\partial f}{\partial z} \\ 2 \times \frac{\partial f}{\partial z} & h\frac{\partial f}{\partial z} - I \end{bmatrix} \tag{27}$$

We determine the eigenvalue of $J$ by solving the equation

$$\det(J - \lambda I) = \begin{bmatrix} h\frac{\partial f}{\partial z} + (1 - \lambda)I & \frac{h^2}{2}\frac{\partial f}{\partial z} \\ 2 \times \frac{\partial f}{\partial z} & h\frac{\partial f}{\partial z} - (1 + \lambda)I \end{bmatrix} = 0 \tag{28}$$

It's trivial to check $J$ satisfies conditions for Lemma A.1.1. Therefore, we have

$$\det(J - \lambda I) = \det\left[\left(h\frac{\partial f}{\partial z} + (1 - \lambda)I\right)\left(h\frac{\partial f}{\partial z} - (1 + \lambda)I\right) - \left(\frac{h^2}{2}\frac{\partial f}{\partial z}\right)\left(2 \times \frac{\partial f}{\partial z}\right)\right] \tag{29}$$

$$= \det\left[-2\lambda h\frac{\partial f}{\partial z} + (\lambda^2 - 1)I\right] \tag{30}$$

Suppose the eigen-decompostion of $\frac{\partial f}{\partial z}$ can be written as

$$\frac{\partial f}{\partial z} = \Lambda \begin{bmatrix} \sigma_1 & & & \\ & \sigma_2 & & \\ & & ... & \\ & & & \sigma_N \end{bmatrix} \Lambda^{-1} \tag{31}$$

Note that $I = \Lambda I \lambda^{-1}$, hence we have

$$\det(J - \lambda I) = \det \Lambda\left\{ -2\lambda h \begin{bmatrix} \sigma_1 & & & \\ & \sigma_2 & & \\ & & ... & \\ & & & \sigma_N \end{bmatrix} + (\lambda^2 - 1)I \right\} \Lambda^{-1} \tag{32}$$

$$= \prod_{i=1}^{N}(\lambda^2 - 2h\sigma_i\lambda - 1) \tag{33}$$

Hence the eigenvalues are

$$\lambda_{i\pm} = h\sigma_i \pm \sqrt{h^2\sigma_i^2 + 1} \tag{34}$$

A-stability requires $|\lambda_{i\pm}| < 1, \forall i$, and has no solution.

The critical boundary is $|\lambda_{i\pm}| = 1$, the solution is: $h\sigma_i$ is 0 or on the imaginary line with norm no larger than 1. □

## A.5 DAMPED ALF

---

**Algorithm 2:** Forward of $\psi$ in Damped ALF ($\eta \in (0, 1]$ )

---

**Input** $(\widehat{z_{in}}, \widehat{v_{in}}, s_{in}, h) = (\widehat{z_0}, \widehat{v_0}, s_0, h)$ where $s_0$ is current time, $\widehat{z_0}$ and $\widehat{v_0}$ are correponding values at time $s_0$; stepsize $h$.

**Forward**

$$s_1 = s_0 + h/2 \tag{35}$$
$$\widehat{z_1} = \widehat{z_0} + \widehat{v_0} \times h/2 \tag{36}$$
$$\widehat{v_1} = f(\widehat{z_1}, s_1) \tag{37}$$
$$\widehat{v_2} = \widehat{v_0} + 2\eta(\widehat{v_1} - \widehat{v_0}) \tag{38}$$
$$\widehat{z_2} = \widehat{z_1} + \widehat{v_2} \times h/2 \tag{39}$$
$$s_2 = s_1 + h/2 \tag{40}$$

**Output** $(\widehat{z_{out}}, \widehat{v_{out}}, s_{out}, h) = (\widehat{z_2}, \widehat{v_2}, s_2, h)$

---

**Algorithm 3:** $\psi^{-1}$ (Inverse of $\psi$) in Damped ALF ($\eta \in (0, 1]$ )

---

**Input** $(\widehat{z_{out}}, \widehat{v_{out}}, s_{out}, h)$ where $s_{out}$ is current time, $\widehat{z_{out}}$ and $\widehat{v_{out}}$ are corresponding values at $s_{out}$, $h$ is stepsize.

**Inverse**

$$(\widehat{z_2}, \widehat{v_2}, s_2, h) = (\widehat{z_{out}}, \widehat{v_{out}}, s_{out}, h) \tag{41}$$
$$s_1 = s_2 - h/2 \tag{42}$$
$$\widehat{z_1} = z_2 - \widehat{v_2} \times h/2 \tag{43}$$
$$\widehat{v_1} = f(\widehat{z_1}, s_1) \tag{44}$$
$$\widehat{v_0} = (\widehat{v_2} - 2\eta\widehat{v_1})/(1 - 2\eta) \tag{45}$$
$$\widehat{z_0} = \widehat{z_1} - \widehat{v_0} \times h/2 \tag{46}$$
$$s_0 = s_1 - h/2 \tag{47}$$

**Output** $(\widehat{z_{in}}, \widehat{v_{in}}, s_{in}, h) = (\widehat{z_0}, \widehat{v_0}, s_0, h)$

---

The main difference between ALF and Damped ALF is marked in blue in Algo. 2. In ALF, the update of $\widehat{v_2}$ is $\widehat{v_2} = (\widehat{v_1} - \widehat{v_0}) + \widehat{v_1} = 2(\widehat{v_1} - \widehat{v_0}) + \widehat{v_0}$; while in Damped ALF, the update is scaled by a factor $\eta$ between 0 and 1, so the update is $\widehat{v_2} = 2\eta(\widehat{v_1} - \widehat{v_0}) + \widehat{v_0}$. When $\eta = 1$, Damped ALF reduces to ALF.

Similar to Sec. A.1, we can write the forward as For simplicity, we can re-write the forward of ALF as

$$\begin{bmatrix} \widehat{z_2} \\ \widehat{v_2} \end{bmatrix} = \begin{bmatrix} \widehat{z_0} + \eta h f(\widehat{z_0} + \frac{h}{2}\widehat{v_0}, s_0 + \frac{h}{2}) + (1-\eta)h\widehat{v_0} \\ 2\eta f(\widehat{z_0} + \frac{h}{2}\widehat{v_0}, s_0 + \frac{h}{2}) + (1-2\eta)\widehat{v_0} \end{bmatrix} \tag{48}$$

Similarly, the inverse of ALF can be written as

$$\begin{bmatrix} \widehat{z_0} \\ \widehat{v_0} \end{bmatrix} = \begin{bmatrix} \widehat{z_2} - h\frac{1-\eta}{1-2\eta}\widehat{v_2} + h\frac{\eta}{1-2\eta}f(\widehat{z_2} - \frac{h}{2}\widehat{v_2}, s_2 - \frac{h}{2}) \\ \frac{1}{1-2\eta}\widehat{v_2} - \frac{2\eta}{1-2\eta}f(\widehat{z_2} - \frac{h}{2}\widehat{v_2}, s_2 - \frac{h}{2}) \end{bmatrix} \tag{49}$$

**Theorem A.3.** *For a single step in Damped ALF with stepsize $h$, the local truncation error of $z$ is $O(h^2)$, and the local truncation errof of $v$ is $O(h)$.*

*Proof.* The proof is similar to Thm. A.3. By similar calculations using the Taylor Expansion in Eq. 15 and Eq. 14, we have

$$
\widehat{z_2} - \tilde{z}(s_0 + h) = (1 - \eta)h\widehat{v_0} + h\eta\Big[f(\widehat{z_0}, s_0) + \frac{h}{2}f_t(\widehat{z_0}, s_0) + \frac{h\widehat{v_0}}{2}f_z(\widehat{z_0}, s_0)\Big]
$$
$$
- h\Big[f(\widehat{z_0}, s_0) + \frac{h}{2}f_t\widehat{z_0}, s_0 + \frac{h}{2}f_z(\widehat{z_0}, s_0)f(\widehat{z_0}, s_0)\Big] + O(h^2) \tag{50}
$$
$$
= (1 - \eta)h\Big(\widehat{v_0} - f(\widehat{z_0}, s_0)\Big) + \frac{\eta - 1}{2}h^2 f_t(\widehat{z_0}, s_0)
$$
$$
+ \frac{h^2}{2}\Big(\eta\widehat{v_0} - f(\widehat{z_0}, s_0)\Big)f_z(\widehat{z_0}, s_0) + O(h^2) \tag{51}
$$

Using Eq. 21, Eq. 15 and Eq. 14, we have

$$
\tilde{v}_2 - \widehat{v_2} = (1 - 2\eta)\widehat{v_0} + (2\eta - 1)f(\widehat{z_0}, s_0) + (1 - \eta)hf_t(\widehat{z_0}, s_0)
$$
$$
+ \Big(\tilde{z}(s_0 + h) - \widehat{z_0} - \eta h\widehat{v_0}\Big)f_z(\widehat{z_0}, s_0) + O(h^2) \tag{52}
$$
$$
= (2\eta - 1)\big[f(\widehat{z_0}, s_0) - \widehat{z_0}\big] + (1 - \eta)hf_t(\widehat{z_0}, s_0)
$$
$$
+ \eta\Big[hf(\widehat{z_0}, s_0) - h\widehat{v_0}\Big]f_z(\widehat{z_0}, s_0) + O(h^2) \tag{53}
$$

Note that when $\eta = 1$, Eq. 51 reduces to Eq. 19, and Eq. 53 reduces to Eq. 26. By initialization, we have $|f(\widehat{z_0}, s_0) - \widehat{v_0}| = 0$ at initial time, hence by induction, the local truncation error for $z$ is $O(h^2)$; the local truncation error for $v$ is $O(h)$ when $\eta < 1$, and is $O(h^2)$ when $\eta = 1$. □

**Theorem A.4** (Theorem 3.2 in the main paper). *For Dampled ALF integrator with stepsize $h$, where $\sigma_i$ is the $i$-th eigenvalue of the Jacobian $\frac{\partial f}{\partial z}$, then the solver is A-stable if $\Big|1 + \eta(h\sigma - 1) \pm \sqrt{\eta\big[2h\sigma_i + \eta(h\sigma_i - 1)^2\big]}\Big| < 1$, $\forall i$.*

*Proof.* The Jacobian of the forward-pass of a single step damped ALF is

$$
J = \begin{bmatrix} I + \eta h\frac{\partial f}{\partial z} & (1 - \eta)hI + \eta\frac{h^2}{2}\frac{\partial f}{\partial z} \\ 2\eta\frac{\partial f}{\partial z} & \eta h\frac{\partial f}{\partial z} + (1 - 2\eta)I \end{bmatrix} \tag{54}
$$

when $\eta = 1$, $J$ reduces to Eq. 27. We can determine the eigenvalue of $J$ using similar techniques. Assume the eigenvalues for $\frac{\partial f}{\partial z}$ are $\{\sigma_i\}$, then we have

$$
\det(J - \lambda I) = \det\begin{bmatrix} (1 - \lambda)I + \eta h\frac{\partial f}{\partial z} & (1 - \eta)hI + \eta\frac{h^2}{2}\frac{\partial f}{\partial z} \\ 2\eta\frac{\partial f}{\partial z} & \eta h\frac{\partial f}{\partial z} + (1 - 2\eta - \lambda)I \end{bmatrix} \tag{55}
$$
$$
= \det\Big[\Big((1 - \lambda)I + \eta h\frac{\partial f}{\partial z}\Big)\Big(\eta h\frac{\partial f}{\partial z} + (1 - 2\eta - \lambda)I\Big)
$$
$$
- \Big((1 - \eta)hI + \eta\frac{h^2}{2}\frac{\partial f}{\partial z}\Big)\Big(2\eta\frac{\partial f}{\partial z}\Big)\Big] \tag{56}
$$
$$
= \prod_{i=1}^{N}\Big[1 + \eta(h\sigma_i - 1) \pm \sqrt{\eta\big[2h\sigma_i + \eta(h\sigma_i - 1)^2\big]}\Big] \tag{57}
$$

when $\eta < 1$, it's easy to check that $\Big|1 + \eta(h\sigma_i - 1) \pm \sqrt{\eta\big[2h\sigma_i + \eta(h\sigma_i - 1)^2\big]}\Big| < 1$ has non-empty solutions for $h\sigma$. □

For a quick validation, we plot the region of A-stability on the imaginary plane for a single eigenvalue in Fig. 1. As $\eta$ increases, the area of stability decreases. When $\eta = 1$, the system is no-where A-stable, and the boundary for A-stability is on the imaginary axis $[-i, i]$ where $i$ is the imaginary unit.

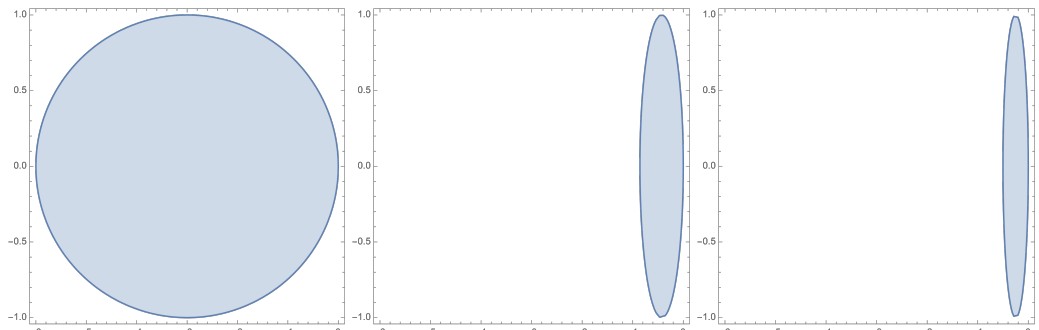

Figure 1: Region of A-stability for eigenvalue on the imaginary plane for damped ALF. From left to right, the region of stability for $\eta = 0.25$, $\eta = 0.7$, $\eta = 0.8$ respectively. As $\eta$ increases to 1, the area of stability region decreases.

## B  EXPERIMENTAL DETAILS

### B.1  IMAGE RECOGNITION

#### B.1.1  EXPERIMENT ON CIFAR10

We directly modify a ResNet18 into a Neural ODE, where the forward of a residual block ($y = x + f(x)$) and the forward of an ODE block ($y = x + \int_0^T f(z, t)dt$ where $T = 1$) share the same parameterization $f$, hence they have the same number of parameters. Our experiment is based on the official implementation by Zhuang et al. (2020) and an open-source repository (Liu, 2017).

All models are trained with SGD optimizer for 90 epochs, with an initial learning rate of 0.01, and decayed by a factor of 10 at 30th epoch and 60th epoch respectively. Training scheme is the same for all models (ResNet, Neural ODE trained with adjoint, naive, ACA and MALI). For ACA, we follow the settings in (Zhuang et al., 2020) and use the official implementation *torch_ACA* [1], and use a Heun-Euler solver with $rtol = 10^{-1}, atol = 10^{-2}$ during training. For MALI, we use an adaptive version and set $rtol = 10^{-1}, atol = 10^{-2}$. For the naive and adjoint method, we use the default *Dopri5* solver from the *torchdiffeq* [2] package with rtol = atol = $10^{-5}$. We train all models for 5 independent runs, and report the mean and standard deviation across runs.

#### B.1.2  EXPERIMENTS ON IMAGENET

**Training scheme**    We conduct experiments on ImageNet with ResNet18 and Neural-ODE18. All models are trained on 4 GTX-1080Ti GPUs with a batchsize of 256. All models are trained for 80 epochs, with an initial learning rate of 0.1, and decayed by a factor of 10 at 30th and 60th epoch. Note that due to the large size input $256 \times 256$, the naive method and ACA requires a huge memory, and is infeasible to train. MALI and the adjoint method requires a constant memory hence is suitable for large-scale experiments. For both MALI and the adjoint menthod, we use a fixed stepsize of 0.25, and integrates from 0 to $T = 1$. As shown in Table. 2 in the main paper, a stepsize of 0.25 is sufficiently small to train a meaningful continuous model that is robust to discretization scheme.

**Invariance to discretization scheme**    To test the influence of discretization scheme, we test our Neural ODE with different solvers *without* re-training. For fixed-stepsize solvers, we tested various step sizes including $\{0.1, 0.15, 0.25, 0.5, 1.0\}$; for adaptive solvers, we set rtol=0.1, atol=0.01 for MALI and Heun-Euler method, and set $rtol = 10^{-2}, atol = 10^{-3}$ for RK23 solver, and set rtol = $10^{-4}, atol = 10^{-5}$ for Dopri5 solver. As shown in Table. 2, Neural ODE trained with MALI is robust to discretization scheme, and MALI significantly outperforms the adjoint method in terms of accuracy (70% v.s. 63% top-1 accuracy on the validation dataset). An interesting finding is that when trained with MALI which is a second-order solver, and tested with higher-order solver (e.g.

---

[1] https://github.com/juntang-zhuang/torch_ACA
[2] https://github.com/rtqichen/torchdiffeq

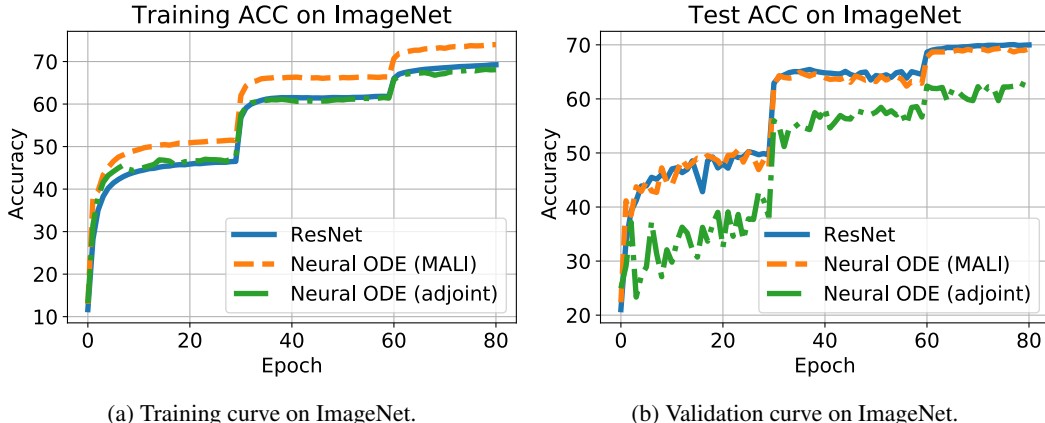

(a) Training curve on ImageNet.

(b) Validation curve on ImageNet.

Figure 2: Results on ImageNet.

RK4), our Neural ODE achieves 70.21% top-1 accuracy, which is higher than both the same solver during training (MALI, 69.59% accuracy) and the ResNet18 (70.09% accuracy).

Furthermore, many papers claim ResNet to be an approximation for an ODE (Lu et al., 2018). However, Queiruga et al. (2020) argues that many numerical discretizations fail to be meaningful dynamical systems, while our experiments demonstrate that our model is continuous hence invariant to discretization schemes.

**Adversarial robustness**   Besides the high accuracy and robustness to discretization scheme, another advantage of Neural ODE is the robustness to adversarial attack. The adversary robustness of Neural ODE is extensively studied in (Hanshu et al., 2019), but not only validated on small-scale datasets such as Cifar10. To our knowledge, our method is the first to enable effectue training of Neural ODE on large-scale datasets such as ImageNet and achieve a high accuracy, and we are the first to validate the robustness of Neural ODE on ImageNet. We use the *advertorch* [3] toolbox to perform adversarial attack. We test the performance of ResNet and Neural ODE under FGSM attack. To be more convincing, we conduct experiment on the pretrained ResNet18 provided by the official PyTorch website [4]. Since Neural ODE is invariant to discretization scheme, it's possible to derive the gradient for attack using one ODE solver, and inference on the perturbed image using another solver. As summarized in Table. 3, Neural ODE consistently achieves a higher accuracy than ResNet under the same attack.

### B.2    TIME SERIES MODELING

We conduct experiments on Latent-ODE models (Rubanova et al., 2019) and Neural CDE (controlled differential equation) (Kidger et al., 2020a). For all experiments, we use the official implementation, and only replace the solver with MALI. The latent-ODE model is trained on the *Mujoco* dataset processed with code provided by the official implementation, and we experiment with different ratios (10%,20%,50%) of training data as described in (Rubanova et al., 2019). All models are trained for 300 epochs with Adamax optimizer, with an initial learning rate of 0.01 and scaled by 0.999 for each epoch. For the Neural CDE model, for the naive method, ACA and MALI, we perform 5 independent runs and report the mean value and standard deviation; results for the adjoint and seminorm adjoint are from (Kidger et al., 2020a). For Neural CDE, we use MALI with ALF solver with a fixed stepsize of 0.25, and train the model for 100 epochs with an initial learning rate of 0.004.

---

[3]https://github.com/BorealisAI/advertorch
[4]https://pytorch.org/docs/stable/torchvision/models.html

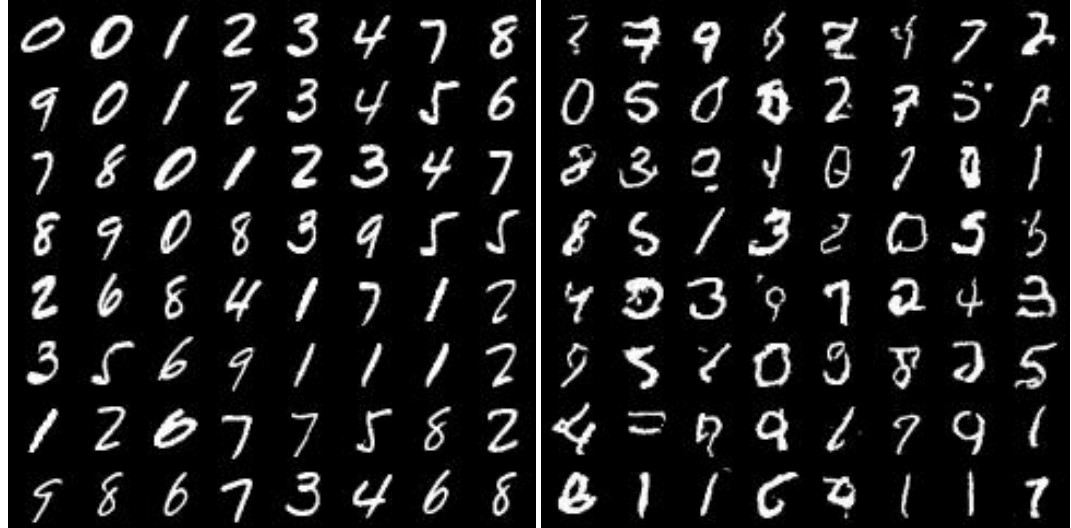

(a) Real samples from MNIST dataset.  (b) Generated samples from FFJORD.

Figure 3: Results on MNIST dataset.

### B.3 CONTINUOUS GENERATIVE MODELS

#### B.3.1 TRAINING DETAILS

Our experiment is based on the official implementation of (Finlay et al., 2020), with the only difference in ODE solver. For a fair comparison, we only use MALI for training, and use *Dopri5* solver from *torchdiffeq* package (Chen et al., 2018) with rtol = atol = $10^{-5}$. For MALI, we use adaptive ALF solver with $rtol = 10^{-2}, atol = 10^{-3}$, and use an initial stepsize of 0.25. Integration time is from 0 to 1.

On MNIST and CIFAR dataset, we set the regularization coefficients for kinetic energy and Frobenius norm of the derivative function as 0.05. We train the model for 50 epochs with an initial learning rate of 0.001.

#### B.3.2 ADDTIONAL RESULTS

We show generated examples on MNIST dataset in Fig. 3, results for Cifar10 dataset in Fig. 4, and results for ImageNet64 in Fig. 5.

### B.4 ERROR IN GRADIENT ESTIMATION FOR TOY EXAMPLES WHEN $t < 1$

We plot the error in gradient estimation for the toy example defined by Eq.6 in the main paper in Fig. 6. Note that the integration time $T$ is set as smaller than 1, while the main paper is larger than 20. We observe the same results, MALI and ACA generate smaller error than the adjoint and the naive method.

### B.5 RESULTS OF DAMPED MALI

For all experiments in the main paper, we set $\eta = 1$ and did not use damping. For completeness, we experimented with damped MALI using different values of $\eta$. As shown in Table. 7, MALI is robust to different $\eta$ values.

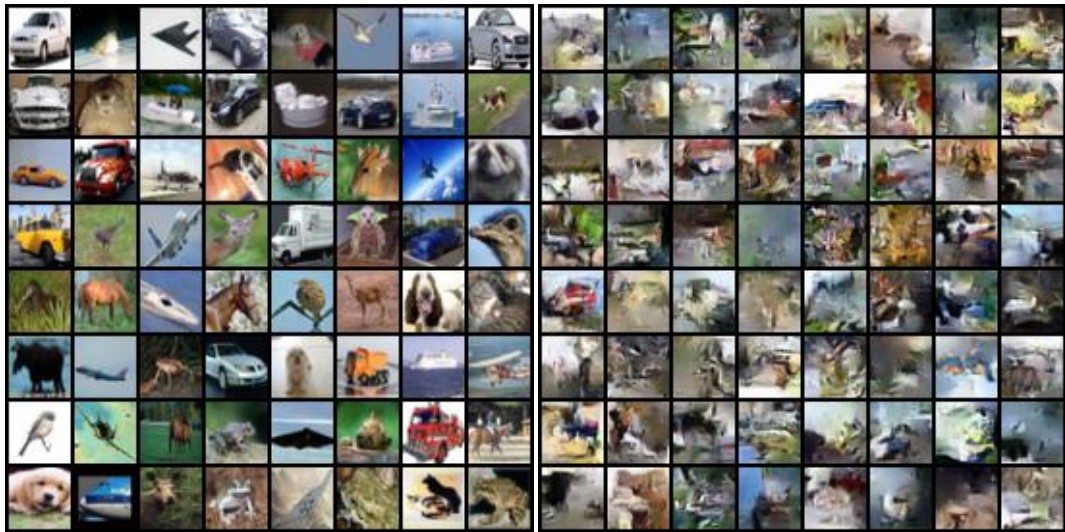

(a) Real samples from CIFAR10 dataset.       (b) Generated samples from FFJORD.

Figure 4: Results on Cifar10 dataset.

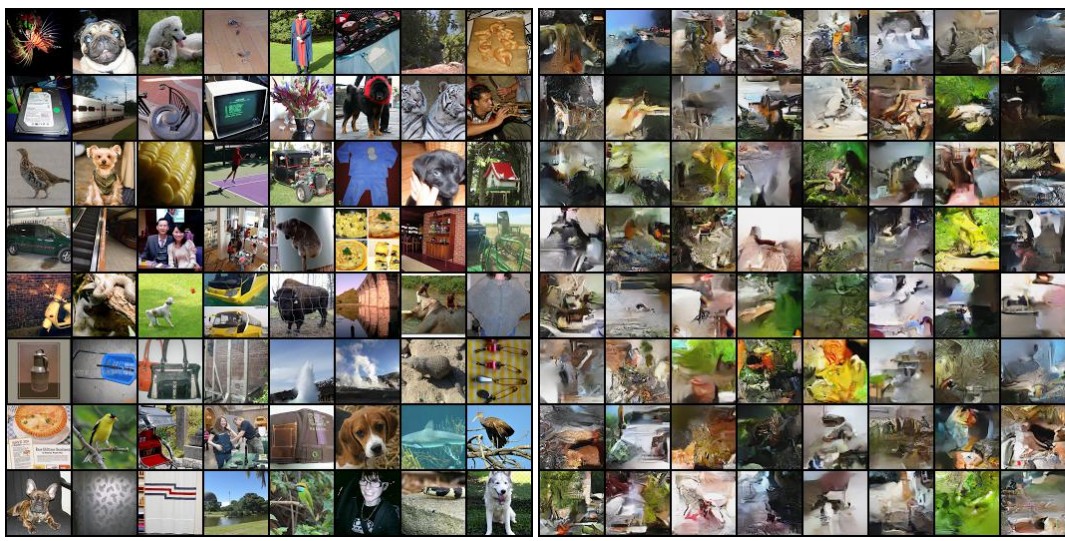

(a) Real samples from ImageNet64 dataset.       (b) Generated samples from FFJORD.

Figure 5: Results on ImageNet64 dataset.

Table 7: Results of damped MALI with different $\eta$ values. We report the test accuracy of Neural CDE on Speech Command dataset, and the test MSE of latent-ODE on Mujoco data.

| $\eta$ | | 1.0 | 0.95 | 0.9 | 0.85 |
|---|---|---|---|---|---|
| Test Accuracy on Speech Commands (Higher is better) | | $93.7 \pm 0.3$ | $93.7 \pm 0.1$ | $93.5 \pm 0.2$ | $93.7 \pm 0.3$ |
| Test MSE of latent ODE | 10% training data | 0.35 | 0.36 | 0.33 | 0.33 |
| on Mujoco (Lower is better) | 20% training data | 0.27 | 0.25 | 0.26 | 0.27 |

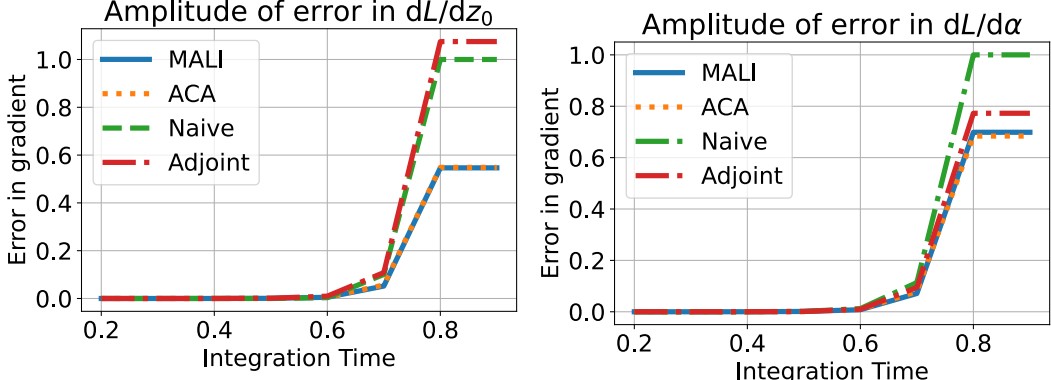

(a) Error in the estimation of gradient *w.r.t* initial condition.

(b) Error in the estimation of gradient *w.r.t* parameter $\alpha$.

Figure 6: Comparison of error in gradient estimation for the toy example by Eq.6 of the main paper, when $t < 1$.

