# OpenReview forum: "MALI: A memory efficient and reverse accurate integrator for Neural ODEs"
_ICLR.cc/2021/Conference — ICLR 2021 Poster_

### Official Review · AnonReviewer1 · 2020-10-25
**Simple method with convincing experimental results, although there are some clarity issues**

**Rating:** 7
**Confidence:** 4

**Review:**

**Paper summary**
There are typically two methods for estimating the gradients with respect to the loss for neural ODEs. The naive method directly backpropagates through the steps of the ODE solver leading to accurate gradients but very large memory cost. The adjoint method in contrast does not store the entire trajectory in memory, but has reverse trajectory errors (i.e. the numerical solution in the reverse direction will not be the inverse of the numerical solution in the forward direction). In this paper, the authors propose a method that is both reverse accurate and has low memory cost.

To achieve this the authors take advantage of the asynchronous leapfrog solver. This numerical method is reversible: solving an ODE numerically in the reverse direction is the inverse of solving the ODE numerically in the forward direction. This is not generally true for the ODE solvers typically used (RK4 and so on) in the neural ODE literature. As the numerical solver is explicitly invertible, the authors can (from only the final state and not the entire trajectory) locally reconstruct each ODE solver step to get the local gradient of the parameters with respect to the loss. They can then calculate these gradients along the entire reverse trajectory to obtain an accurate estimate of the gradient with respect to the parameters. As each step of the numerical solver is reversible, they do not need to store the entire trajectory. The authors analyse the stability and numerical error of their proposed method and provide a toy example to show how well their method estimates gradients compared the naive, adjoint and adaptive checkpoint methods.

The authors then perform experiments on a variety of tasks to test their model. They test their model on image classification experiments, both on CIFAR10 and Imagenet and achieve good results compared to the baselines. In addition, they perform adversarial robustness experiments on ImageNet and also show good performance. Finally, the authors test their method both for time series modeling and continuous normalizing flows, again showing good performance compared with naive integration methods.

**Positives**
- The motivation and core idea of the paper is clear. Numerical solvers are in general not reversible and this can lead to inaccurate gradient estimates when using the adjoint method. The authors explain this clearly and then propose a method that effectively solves this.
- The experimental results are quite impressive. The model performs on par with the adaptive checkpoint method in terms of accuracy but is much more memory efficient (and notably memory is independent of the number of integration steps). This allows the authors to run their model on large scale datasets like ImageNet which was not previously possible with most neural ODE methods. Further, the authors achieve good performance on quite a wide variety of tasks (image classification, adversarial attacks, time series modeling, generative modeling) which is nice.
- The authors perform a thorough analysis of the runtime of their integration method compared to others which is very helpful.

**Negatives**
- The presentation of the method and results is not always very clear. For example, the section about damping for the ALF integrator is not clear. The authors mention that ALF is not stable for eta=1, but (as far as I can tell) never mention what value of eta they use in practice and whether choosing this value is difficult. Further, it is not clear if ALF is still reversible with this eta parameter. Presumably you would have to use 1/eta in the reverse step for it to remain invertible, in which case the reverse is not stable? The authors should be more clear about this.
- The toy example is confusing. How come the integration time starts from t=20? Is this because the error only grows after t=20? As you use T=1 for all experiments (and the rtol and atol are also roughly the same for all experiments), it would be nice to see if this actually makes a difference also for t<1. In Figure 4, the authors also mention the derivative dL/dy_0 but this derivative is never mentioned in the text. Do you mean dL/dz_0? The plots of memory consumption are nice and clear though.
- The ALF solver already exists, so the main contribution of the paper is simply to apply the ALF solver to neural ODEs. This means that the novelty of the method is somewhat limited, but I do not think that this is a major issue as the method works well and is clearly motivated.
- The section about using different numerical solvers for ResNets does not make much sense. ResNets are not designed as flows and do not behave as flows in practice, so we should not expect them to work at all with other numerical solvers than Euler with timestep=1. I don’t really think these experiments show anything interesting and should be removed for clarity.

**Recommendation**
Overall the paper has a clear motivation, provides a nice and simple solution to an interesting problem and has good experimental results. However, there are some clarity issues which make some aspects of the model and method confusing. I therefore recommend a weak accept but would increase my score if the clarity issues are solved.

**Questions**
The model achieves extremely good bits/dim on MNIST (0.87). However, it seems from the appendix that the samples are fairly poor (compared to vanilla FFJORD for example). Log likelihood and sample quality are not always correlated, but the difference seems particularly jarring here. Do you know why this is?

**Typos and small comments**
- In many places the authors use latex math to write words that should either just be written in italics (w.r.t) or using \text{} in math mode (e.g. atol, rtol).
- There are several typos in the script, so I think it would be a good idea for the authors to read through the script again to fix those.
- In several places, the authors write O(N_f + 1) which instead should be O(N_f)
- The authors often write “constant memory cost with respect to integration time”. I think it would be more helpful to say “number of solver steps” or something along those lines as integration time typically refers to the upper limit of the integral when solving an ODE.

---

> ### Author Response · Authors · 2020-11-13
> **Response to reviewer 1**
>
> Thanks for the insightful comments, we address your concerns below.
>
> **Negative 1: the section about damping for the ALF integrator is not clear.**
>
> Thanks for pointing out. For all experiments in current submission, $\eta$ is set as 1. We conducted experiments with $\eta<1$, and found that MALI is typically robust to different $\eta$ values. Link to results: https://openreview.net/forum?id=blfSjHeFM_e&noteId=hmWwoXnCaN
>
> With $\eta<1$, Damped ALF is still invertible, we have updated the inverse of damped ALF in Appendix A.5.
> To consider the stability of inverse of ALF, we need to calculate the eigenvalue of the Jacobian. Note that the Jacobian of inverse ALF is the inverse of the Jacobian for forward ALF. Also note that the eigenvalue of the inverse of a matrix is the inverse of its eigenvalue, if the eigenvalue is not 0. Therefore, stability of inverse ALF requires Eq.34 in Appendix to have a norm larger than 1, and the stability region is non-empty.
>
> **Negative 2: it would be nice to see if this actually makes a difference also for t<1**
>
> Thanks for pointing out, we use T=20 mainly for the purpose for visualization. We added an example when T<1 in Fig.6 of the appendix; MALI achieves comparable error as ACA, and both outperform the naïve and adjoint method.
> Fig. 4 caption has a typo, it should be dL/dz_0; we have updated it in the revision.
>
> **Negative 4: The section about using different numerical solvers for ResNets does not make much sense.**
>
> We have removed this part in the revision as suggested by the reviewer. As stated in Sec4.2 “Invariance to discretization scheme”, we agree with the reviewer and think that “ResNet as a one-step Euler discretization fails to be a meaningful dynamical system”. However, as stated in [1], “residual units are interpreted as one step of a forward Euler method for the numerical integration of a parameterized ordinary differential equation (ODE)”[2,3]. In order to test this hypothesis in the original submission, we summarize results for ResNet with different ODE solvers.
>
> **Question: Results of FFJORD on MNIST**
>
> We apologize that we used the wrong figure for generated examples for the MNIST dataset. The figures for MNIST were generated using a low temperature (~0.7). We have revised the paper to use examples generated with temperature=1.0. With a low temperature, the hidden state is close to 0 and the model tends to generate similar figures (visually the figures under low temperature are more blurry).
>
> We are not aware of the exact reason for the mismatch between sample quality and log probability. But we found reference [4] stated that “In particular, we show that three of the currently most commonly used criteria—average log-likelihood, Parzen window estimates, and visual fidelity of samples—are largely independent of each other when the data is high-dimensional.”
>
> **Typos and other comments**
> We have fixed typos as suggested, and replaced the big O notation with the actual cost (note that $N_f$ is not large in practice), and modified the description of constant memory.
>
> [1] Queiruga, Alejandro F., et al. "Continuous-in-Depth Neural Networks." arXiv preprint arXiv:2008.02389 (2020).
>
> [2] Eldad Haber and Lars Ruthotto. Stable architectures for deep neural networks.Inverse Problems, 34(1):014004,2017.
>
> [3] Eldad Haber, Keegan Lensink, Eran Treister, and Lars Ruthotto. IMEXnet: A forward stable deep neural network.InInternational Conference on Machine Learning, pages 2525–2534, 2019.
>
> [4] Lucas Theis, Aäron van den Oord, and Matthias Bethge. A note on the evaluation of generative models. arXiv preprint arXiv:1511.01844, 2015.

---

> > ### Comment · AnonReviewer1 · 2020-11-20
> > **Thank you for your response**
> >
> > Thank you for addressing most of the concerns I brought up.
> > 1. I appreciate the updated appendix and discussion on this. The experiments with `$\eta < 1$ are also convincing.
> > 2. Thank you for updating this. The figure in the appendix is a little surprising as the error seems to become flat for all models after $t > 0.8$. However, it does show that the claims are true for $T=1$ as well.
> > 3. Thank you for removing this. I think it makes the paper a lot clearer.
> >
> > Thank you also for updating the MNIST samples image. As mentioned in the review, sample quality and log likelihood are not always correlated (as noted in the Theis paper). However, it is surprising that for the *same* model (FFJORD) with better training and better log likelihood, the sample quality decreases. Either way, I thank the reviewers for updating the model samples and hopefully some of these differences can be explained with future research.
> >
> > As the authors have addressed most of my concerns, I have updated my score from 6 to 7 and think this paper deserves to be accepted at ICLR.

---

> > > ### Author Response · Authors · 2020-11-21
> > > **Thanks for your review**
> > >
> > > Thanks for reviewing our paper and author response, please let us know if you have any further questions.

---

### Official Review · AnonReviewer3 · 2020-10-27
**Gradient estimation of neural ODEs with constant memory footprint analysed and compared**

**Rating:** 6
**Confidence:** 2

**Review:**

1) Summary
The manuscript proposes a reversible integration scheme for approximately estimating the gradient of neural ordinary differential equations. These ODEs represent DNNs with continuous rather than discrete values for the number of layers. The solver is theoretically analysed and empirically compared to other solvers.

2) Strengths
+ The paper is mostly well written.
+ The reversibility property of the solver leads to a memory footprint that does not depend on integration time.
+ The model is applied to standard datasets.

3) Concerns
- The concept of Neural ODE models, their scope and their expected usefulness should be better motivated. It is not obvious which role these models play and what they offer as potential strengths.
- The integrations scheme seems already known and well established.
- It does not seem that the paper makes code and data available to the public.

4) Remarks/Questions
  a) Algorithm 1: It seems that the stepsize h should be initialized upon every step. Otherwise the steps can only get smaller.
  b) References: capitalization not correct e.g. "neural information processing systems", "ode-net", "lennard-jones"
  c) What benefits does the neural ODE model have in the context of image classification? What is the intuition behind the "continuous depth" idea in this scenario?

---

> ### Author Response · Authors · 2020-11-13
> **Response to reviewer 3**
>
> Thanks for the nice review, we address your concerns below.
>
> **Concern 1: The concept of Neural ODE models, their scope and their expected usefulness should be better motivated.**
>
> Thanks for pointing out. The motivation for Neural ODE is briefly discussed in the introduction, but not explained in detail due to the limited paper length. We summarize a few motivations here.
>
> Conventional RNNs typically require the data to be sampled in an even grid, so that the time step between two consecutive observations is a constant. However, in real-world applications, such as tracking the record of a patient, the data is often irregularly sampled. A continuous ODE model can be evaluated at arbitrary time points, hence is suitable for this case [3].
>
> Much knowledge about physics and biology are in the form of differential equations. Hence it’s natural to incorporate such knowledge into a Neural ODE model, such as the Hamiltonian system [1] or three-body system [2].
>
> Neural ODEs can also be used for generative modeling. Discrete-layer normalizing flows typically have special structures to guarantee invertibility, while continuous ODE models are naturally invertible (in theory) hence is suitable for free-form generative models [4].
>
> **Concern 2: The integrations scheme seems already known and well established.**
>
> The method of ALF is published on 2011 [5], however it’s a pity that it’s never cited or noticed. We found it to be perfectly suitable for the high-dimensional Neural ODE models.
>
> **Concern 3: It does not seem that the paper makes code and data available to the public.**
>
> Sorry we were not able to polish the code before submission, we provide link to our code https://www.dropbox.com/s/8tu0lck9ftpve4n/code_for_MALI.zip?dl=0
>
> **Remark a):It seems that the stepsize h should be initialized upon every step. Otherwise the steps can only get smaller**
>
> Thanks for pointing out. We found some typo in Algo 1. If the error estimate is below tolerance, then the stepsize is increased by a certain factor; if the error estimate is above tolerance, then stepsize is decreased. We updated this in the revision.
>
> **Remark b): References: capitalization not correct**  Thanks for pointing out, we have fixed these in the revision.
>
> **Remark c): What benefits does the neural ODE model have in the context of image classification**
>
> For image classification, the biggest advantage of Neural ODE model is their robustness to adversarial attacks. This is extensively studied in [6]. The intuition is ODE models with bounded norm on the gradient (this is often satisfied in Neural ODE since the weights can only take finite real values in popular frameworks such as PyTorch and Tensorflow) defines a bijection between the initial state and the end-time state. In other words, the integral curves are non-intersecting [6]. In contrast for discrete models such as ResNet, the curves can easily intersect, hence small perturbations in the input can cause very different output.
>
> [1] Desmond Zhong, Yaofeng, Biswadip Dey, and Amit Chakraborty. "Symplectic ODE-Net: Learning Hamiltonian Dynamics with Control." arXiv (2019): arXiv-1909.
>
> [2] Zhuang, Juntang, et al. "Adaptive Checkpoint Adjoint Method for Gradient Estimation in Neural ODE." arXiv preprint arXiv:2006.02493 (2020).
>
> [3] Rubanova, Yulia, Ricky TQ Chen, and David Duvenaud. "Latent odes for irregularly-sampled time series." arXiv preprint arXiv:1907.03907 (2019).
>
> [4] Grathwohl, Will, et al. "Ffjord: Free-form continuous dynamics for scalable reversible generative models." arXiv preprint arXiv:1810.01367 (2018).
>
> [5] Mutze, Ulrich. "An asynchronous leapfrog method II." arXiv preprint arXiv:1311.6602 (2011).
>
> [6] Hanshu, Y. A. N., et al. "On robustness of neural ordinary differential equations." International Conference on Learning Representations. 2019.

---

### Official Review · AnonReviewer2 · 2020-10-28
**A good submission & a few questions**

**Rating:** 7
**Confidence:** 3

**Review:**

Summary: This paper presents a memory-efficient asynchronous leapfrog integrator for numerically solving neural ODEs, referred to as MALI. The method comes with a constant memory guarantee (like the adjoint method) and also guarantees reverse-time accuracy (like the adaptive checkpoint adjoint (ACA) method). The authors also give a rigorous theoretical analysis of MALI, and also discuss a "damped" version with an increased stability region. The method is evaluated on a variety of tasks which includes classification, dynamical modelling and generative modelling.

Pros:
- The theoretical analysis looks correct, noting that I haven't worked out all the details.
- Experimental evaluation is very exhaustive, and MALI achieves near-the-best performance in all tasks.
- The method is proven as accurate as the standard numerical ODE solvers. Thanks to its reduced memory cost (compared to ACA), MALI can then be treated as an off-the-shelf replacement.

Cons and Questions:
- Looking at the results, I'm having difficulty seeing any significant improvement upon ACA. Then the main contribution (in addition to the theoretical analysis) is the reduced memory consumption, which makes me rethink whether ICLR is a suitable venue.
- Although the memory costs of the adjoint method and MALI are $O(N_f)$ and $O(N_f+1)$, this doesn't really reflect in Figure 4c, where the blue bar doubles the red one. I'd be happy if the authors can briefly explain why
- Looking at Table-2, why does the test performance of a NODE trained with MALI increase when we switch from MALI to RK4? It would be much nicer to see some error variance estimate.
- I would be happy to see an experimental evaluation of the "A-stability". As mentioned by the authors, the stability analysis is asymptotic and T could be arbitrarily small in, e.g., continuous-time flows. However, that's not the case in time-series modelling. So I wonder if the stability claim can be verified on a scenario in which, e.g., 100 observations arrive uniformly in time with 10 secs gaps.
- To generate Table-2, did you train a ResNet without any differentials/integrals involved and try to evaluate the test performance using an ODE solver (simply using the trained ResNet as the drift function)? If so, I don't think this makes any sense except Euler-1 solver, and the entire ResNet row in Table-2 could go away.

Additional comments:
- Figure-4 caption could include some more detail (at least mentioning the experiment name)
- Why is there a "local forward" step within the for loop in the backward routine in Alg.4?
- It would be nice to see a brief description of the mujoco dataset.
- Typo in the title of section B.3.2.

Note: After rebuttal, I increase my overall score from 6 to 7.

---

> ### Author Response · Authors · 2020-11-13
> **Response to reviewer 2**
>
> Thanks for the insightful comments, and we address your concerns below.
>
> **Con 1: Looking at the results, I'm having difficulty seeing any significant improvement upon ACA.**
>
> The biggest advantage of MALI over ACA is in the constant memory cost, hence is suitable for high-dimensional problems. The memory issue is critical in practice; note ACA cannot run on continuous normalizing flow even on Cifar10 because of its large memory, and ACA cannot deal with large datasets like ImageNet.
>
> **Con 2: this doesn't really reflect in Figure 4c, where the blue bar doubles the red one. I'd be happy if the authors can briefly explain why**
>
> We have updated table 1 and remove the big $O$ notation. Using $N_z$ to denote the dimension of hidden state $z$, the memory for adjoint is $N_z N_f$, while is $N_z (N_f+1)$ for MALI.
>
> Note that $N_f$ is not a large number, in Fig 4(c) we use a two-layer network in $f$, hence $N_f$=2 and $N_f+1=3$, and the ratio $(N_f+1)/N_f$ is 1.5.
>
> Furthermore, we record the peak memory usage, note that in Algorithm 2, there are extra variables $k_1$ and $u_1$, they also count toward the “peak memory” usage, and they are deleted after the entire execution of Algo 2 (rather than immediately deleted after usage) in coding, hence this contributes to the “peak memory”. However, the code can be easily modified to delete them right away after usage in order to keep a low peak memory usage.
>
> **Con 3: why does the test performance of a NODE trained with MALI increase when we switch from MALI to RK4?**
>
> The increase in test accuracy perhaps is because our method is actually learning a continuous model, hence is invariant to different numerical ODE solvers. Note that RK4 is typically more accurate than MALI: for a stepsize h, the local error of RK4 is of order $O(h^4)$, while is $O(h^3)$ for MALI, hence RK4 is perhaps more accurate in terms of numerically approximating an ODE.
>
> Due to limited computation resource, we were not able to perform several experiments on ImageNet to obtain variance estimates. We conducted 5 independent experiments on CIFAR10, and Fig. 5 shows some variance estimates for the accuracy of different methods.
>
> **Con 4: I would be happy to see an experimental evaluation of the "A-stability".**
>
> For all experiments in current submission, we set $\eta=1$. We conducted extra experiments with $\eta<1$, and MALI is robust to different choices of $\eta$. Note that the latent-ODE experiment is performed on the Mujoco dataset, with each time series has about 100 observation time points.  Link to results: https://openreview.net/forum?id=blfSjHeFM_e&noteId=hmWwoXnCaN
>
> **Con 5: To generate Table-2, did you train a ResNet without any differentials/integrals involved and try to evaluate the test performance using an ODE solver**
>
> The ResNet is trained without considering integration, and we have removed this part in revision as suggested.
> As stated in Sec4.2 “Invariance to discretization scheme”, we agree with the reviewer and think that “ResNet as a one-step Euler discretization fails to be a meaningful dynamical system”. However, many papers perform theoretical analysis of ResNet as an approximation to ODE. In order to test this hypothesis, we summarize results for ResNet with different ODE solvers, and mark the result for one-step Euler solver in bold font to emphasize the best result of ResNet.
>
> **Additional comment 1: Caption of Fig.4**
> Thanks for pointing out, we have updated in the revision.
>
> **Additional comment 2: Why is there a "local forward" step within the for loop in the backward routine in Alg.4?**
>
> The computation graph associated with “local forward” is deleted during forward-time integration to save memory. Hence in the backward pass, we need to reconstruct the local forward to calculate the gradient. The same is required for the adjoint method in the “torchdiffeq” package accompanying [1]; see local forward: https://github.com/rtqichen/torchdiffeq/blob/5dbaf8585ff9e601889811b7a5859ccf87dc576a/torchdiffeq/_impl/adjoint.py#L98,
> local backward: https://github.com/rtqichen/torchdiffeq/blob/5dbaf8585ff9e601889811b7a5859ccf87dc576a/torchdiffeq/_impl/adjoint.py#L105.
>
> **Additional comment 3: It would be nice to see a brief description of the mujoco dataset.**
>
> We have added description and reference as suggested.
>
> **Additional comment 4: Typo in the title of section B.3.2.**
>
> Thanks for pointing out, we fixed it in revision.

---

> > ### Comment · AnonReviewer2 · 2020-11-21
> > **Thanks for the response!**
> >
> > I'd like to thank the authors for their well-written and elaborate reply! All my concerns are nicely addressed. Now, Table-1 and Table-2 read much better. Maybe two minor comments:
> >
> > - RK4 performing better than MALI is still mysterious. If, as authors claimes, that was because RK4 having a smaller error due to discretization, then wouldn't an adaptive solver like dopri5 give even superior performance?
> > - Since Table-2 is updated, I believe the paragraph starting with "Furthermore, many papers claim ResNet" should be updated as well.
> >
> > I increase my overall score to 7. Nice contribution!

---

> > > ### Author Response · Authors · 2020-11-23
> > > **Thanks for your review**
> > >
> > > Thanks for the insightful review for our submission and author response.
> > > * We have updated the paragraph as suggested.
> > > *For the experiment of different solvers, it's likely that Dopri5 achieves a lower accuracy because the error tolerance is too large (rtol=1e-2). We conducted another experiment with Dopri5 using rtol=1e-5 and achieved a better test accuracy of 70.07 on ImageNet. Though still slightly worse than the best of RK4, it demonstrates the trend that as error tolerance of adaptive ODE solver decreases, the accuracy increases.
> > >
> > > Thanks for the review. Please let us know if you have any further questions.

---

### Official Review · AnonReviewer4 · 2020-10-29
**Neural ODEs + Invertible networks**

**Rating:** 7
**Confidence:** 2

**Review:**

Summary:
This paper proposes a new algorithm for solving neural ODEs. Each numerical solver step of the neural ODE is implemented as an invertible neural network via a variant of the asynchronous leafprog integrator. While still computing an accurate gradient, this allows memory savings by discarding intermediate data from the numerical integration steps since it can be reconstructed using the inverse. A theoretical stability analysis is provided. The experimental results show that the algorithm achieves similar performance to previous methods (e.g. ACA) while using less memory.


Strengths:
+ Identifies a nice connection between invertibility and memory efficiency. Beyond neural ODEs, this could enable use of larger models where invertible networks are useful (e.g. normalizing flows)
+ The theoretical analysis of stability is useful to build intuition


Concerns / weaknesses:
- Most experiments in the paper use damping_factor $= \eta = 1$. Since theoretically, this is not stable, it would be nice to see if the empirical improvements hold up for $\eta < 1$, where stable regions do exist
- The naive method seems too naive. Why are all results from all $m$ evaluations being saved? It is obvious that $m-1$ of these are unnecessary for gradient computation since they don't affect $z(T)$. The related claim about the computation graph being deeper for the naive method also seems incorrect.


Other comments:
- In Algorithm 1, shouldn’t $error_est = \inf$ be inside the while loop?
- In Algorithm 4, shouldn’t $a(T)$ be the partial derivative of $L$ wrt $z(T)$ instead of total derivative?
- In Theorem 3.2, what is $\sigma$? Should it be $\sigma_i$?
- In various locations, notation like $O(N+1)$ is used. Should this just be $O(N)$ since I assume $N$ is at least $\Omega(1)$?
- It seems a bit strange that we have to do a local forward and local backward pass in Algorithm 4. Could this be solved by making each layer of f invertible? In the same vein, it seems that the adjoint method needs to do a separate solve of the reverse ODE because of loss of information. If we were to assume invertibility of the forward map, is there a way to modify the adjoint method to exactly retrace the path backwards?

---

> ### Author Response · Authors · 2020-11-13
> **Response to reviewer 4**
>
> Thanks for the constructive review. We address your concerns below.
>
> **Concern 1: it would be nice to see if the empirical improvements hold up for $\eta<1$, where stable regions do exist**
>
> For all experiments in current submission, we set $\eta=1$. We conducted extra experiments with $\eta<1$, and MALI is robust to different choices of $\eta$. https://openreview.net/forum?id=blfSjHeFM_e&noteId=hmWwoXnCaN
>
> **Concern 2: The naive method seems too naive. Why are all results from all evaluations being saved?**
>
> The “naive” integration is from the “torchdiffeq” package accompanying [1]; it does not discard unused steps to search for optimal stepsize.
> This can be found at https://github.com/rtqichen/torchdiffeq/blob/5dbaf8585ff9e601889811b7a5859ccf87dc576a/torchdiffeq/_impl/misc.py#L95, where the variable “error_ratio” is  calculated from the current step and is used to estimate the next stepsize. This function is called recursively until an acceptable stepsize is found. Note that “error_ratio” is not detached from the computation graph, hence the extra $m-1$ step is back-propagated in the “odeint” method: https://github.com/rtqichen/torchdiffeq/blob/5dbaf8585ff9e601889811b7a5859ccf87dc576a/torchdiffeq/_impl/odeint.py#L27. (Note “odeint” is not “odeint_adjoint”)
>
> ACA proposed to discard the additional $m-1$ steps [2], and ACA typically achieves better accuracy than the adjoint.
>
> **Other comments 1-3: Fix typos**
>
> Thanks for pointing out the typos in 1-3 of "other comments", we have updated in the revision.
>
> **Other comments 4: In various locations, notation like $O(N+1)$is used. Should this just be $O(N)$ since I assume is $N$ at least $\Omega(1)$?**
>
> We originally use $O(N_f+1)$ in order to emphasize the increased memory cost for MALI compared with the adjoint method. Also note that $N_f$ is typically less than 10 for one ODE block.
> We update the table to reflect actual costs, removing the big $O$ notations and multiplying the memory and computation cost by $N_z$, which is the dimension of the hidden state.
>
> **Other comments 5: It seems a bit strange that we have to do a local forward and local backward pass in Algorithm 4**
>
> The local forward and backward evaluation is necessary to evaluate $\frac{\partial f}{\partial z}$ and $\frac{\partial f}{\partial \theta}$, because it's deleted in the forward-time integration to save memory. It is also necessary for the “adjoint” method in the “torchdiffeq” package.
> Local forward: https://github.com/rtqichen/torchdiffeq/blob/5dbaf8585ff9e601889811b7a5859ccf87dc576a/torchdiffeq/_impl/adjoint.py#L98,
> local backward:  https://github.com/rtqichen/torchdiffeq/blob/5dbaf8585ff9e601889811b7a5859ccf87dc576a/torchdiffeq/_impl/adjoint.py#L105
>
> **Other comments 5: Could this be solved by making each layer of $f$ invertible?**
>
> For general form of $f$ and general ODE solvers (such as the Runge-Kutta method), to our knowledge no existing methods can guarantee invertibility and accurately reconstruct the forward-time trajectory only from the end-time state.
> To our knowledge, MALI (ALF) is the first method to achieve invertibility for general form of $f$.
> It’s possible to use some special invertible structures in $f$, so every evaluation of $f$ is invertible regardless of the ODE solver; however, such structures might be unsuitable for many practical problems. For example, in some biological and physical ODE models, $f$ takes a linear form and  the numerical integration is typically not strictly explicitly invertible. Though there are implicit methods, it typically requires iteration for each step and is not suitable for high-dimensional cases.
>
> [1] Chen, Ricky TQ, et al. "Neural ordinary differential equations." Advances in neural information processing systems. 2018.
>
> [2] Zhuang, Juntang, et al. "Adaptive Checkpoint Adjoint Method for Gradient Estimation in Neural ODE." arXiv preprint arXiv:2006.02493 (2020).

---

### Author Response · Authors · 2020-11-13
**Updated results for $\eta<1$**

All results in the submission are generated using $\eta=1$. We have updated the results for $\eta<1$ as suggested by reviewers. MALI demonstrate robustness to different $\eta$ values.

|                        $ \eta $                               |                         |        1.0         |     0.95    |     0.9     |     0.85     |
|:---------------------------------------------------:|:-----------------:|:-----------:|:-----------:|:-----------:|:------------:|
| Test Accuracy on Speech Commands (Higher is better) |                   | $93.7\pm0.3$ | $93.7\pm0.1$ | $93.5\pm0.2$ | $93.7\pm0.3$ |
| Test MSE of latent ODE  on Mujoco (Lower is better) | 10% training data |     0.35    |     0.36    |     0.33    |     0.33     |
|                            Test MSE of latent ODE  on Mujoco (Lower is better)                          | 20% training data |     0.27    |     0.25    |     0.26    |     0.27     |

---

### Comment · ~Patrick_Kidger1 · 2021-01-16
**Reference Fix**

Nice paper! I'm giving this a try myself.

Just commenting to say that the same "Kidger et al. (2020)" is cited for both neural CDEs (https://arxiv.org/abs/2005.08926) and seminorm adjoints (https://arxiv.org/abs/2009.09457). (The neural CDE reference is the one that is missing.)

---

> ### Author Response · Authors · 2021-01-16
> **Thanks for pointing out, will fix the reference**
>
> Thanks a lot. It might be due to the same abbreviation is used in latex reference. We will fix this in the final version.

---

### Decision · Program_Chairs · 2021-01-07
**Final Decision**

**Decision:**

Accept (Poster)

**Comment:**

This paper introduced a new ODE integration scheme that allows constant-memory gradient computation.  I was concerned that the low order of convergence of this method would make it impractical, but the authors performed extensive experiments and got impressive results.  Overall the paper addresses one of the main practical difficulties with large neural ODE models.  The authors satisfactorily addressed the reviewers' concerns in the discussion.